# Prior Diffusiveness and Regret in the Linear-Gaussian Bandit

**Yifan Zhu** [1]   **John C. Duchi** [1]   **Benjamin Van Roy** [1]

## Abstract

We prove that Thompson sampling exhibits $\widetilde{O}(\sigma d\sqrt{T} + dr\sqrt{\mathrm{Tr}(\Sigma_0)})$ Bayesian regret in the linear-Gaussian bandit with a $\mathcal{N}(\mu_0, \Sigma_0)$ prior distribution on the coefficients, where $d$ is the dimension, $T$ is the time horizon, $r$ is the maximum $\ell_2$ norm of the actions, and $\sigma^2$ is the noise variance. In contrast to existing regret bounds, this shows that to within logarithmic factors, the prior-dependent "burn-in" term $dr\sqrt{\mathrm{Tr}(\Sigma_0)}$ decouples additively from the minimax (long run) regret $\sigma d\sqrt{T}$. Previous regret bounds exhibit a multiplicative dependence on these terms. We establish these results via a new "elliptical potential" lemma, and also provide a lower bound indicating that the burn-in term is unavoidable.

## 1. Introduction

In the linear-Gaussian bandit, at times $t = 0, 1, 2, \ldots$, a player sequentially chooses an action $A_t \in \mathcal{A} \subset \mathbb{R}^d$ and receives a reward

$$R_{t+1} = \theta_\star^\top A_t + W_{t+1}, \quad W_{t+1} \stackrel{\mathrm{iid}}{\sim} \mathcal{N}(0, \sigma^2), \quad (1)$$

where $\mathcal{A}$ denotes the action set and $\theta_\star$ an unknown vector, and the goal is to maximize the rewards. Thompson, or posterior, sampling (Thompson, 1933) has proven surprisingly (empirically) effective for this problem. It proceeds by first placing a prior distribution $\rho$ on $\theta_\star$. At time $t$, conditional on the history

$$H_t := (A_0, R_1, \ldots, A_{t-1}, R_t)$$

of actions and rewards to time $t$, Thompson sampling draws

$$\hat{\theta}_t \sim \rho(\cdot \mid H_t),$$

[1]Department of Electrical Engineering, Stanford University, California, United States. Correspondence to: Yifan Zhu <zhuyifan@stanford.edu>.

*Proceedings of the 43rd International Conference on Machine Learning*, Seoul, South Korea. PMLR 306, 2026. Copyright 2026 by the author(s).

that is, a vector from the posterior on $\theta_\star$ given $H_t$, and then takes the (putatively) optimal action conditional on $\hat{\theta}_t$ via

$$a_\star(\theta) \in \arg\max_{a \in \mathcal{A}} \theta^\top a, \quad \text{i.e.} \quad A_t = a_\star(\hat{\theta}_t). \quad (2)$$

An optimal action is $A_\star := a_\star(\theta_\star)$, and in this framework, we study the Bayesian regret

$$\mathrm{Reg}(T) := \sum_{t=0}^{T-1} \mathbb{E}\left[\theta_\star^\top A_\star - R_{t+1}\right], \quad (3)$$

where the expectation is taken over the random $\theta_\star \sim \rho$, the random noise $W_t$, and any randomness in the actions chosen.

When the prior $\rho$ is Gaussian, so that $\theta_\star \sim \mathcal{N}(0, \Sigma_0)$, letting $\boldsymbol{A}_t = [A_0 \cdots A_{t-1}]^\top \in \mathbb{R}^{t \times d}$ be the matrix of actions and $\boldsymbol{R}_t = [R_1 \cdots R_t]^\top$ the vector of rewards, the posterior for the linear model (1) takes a particularly simple form. Indeed, for

$$\Sigma_t = \left(\sigma^{-2}\boldsymbol{A}_t^\top \boldsymbol{A}_t + \Sigma_0^{-1}\right)^{-1},$$

we have

$$\theta_\star \mid H_t \sim \mathcal{N}\left(\sigma^{-2}\Sigma_t \boldsymbol{A}_t^\top \boldsymbol{R}_t, \Sigma_t\right),$$

so that sampling $\hat{\theta}_t$ is easy, and typically (e.g., if $\mathcal{A}$ is a scaled $\ell_2$-ball), so is sampling the actions (2).

The "canonical" linear-Gaussian bandit takes the action set $\mathcal{A} \subset r\mathbb{B}_2^d$, the $\ell_2$-ball of radius $r$. For this canonical setting, Kalkanli & Özgür (2020) provide what we consider a proto-typical regret bound, developing an information-theoretic analysis to demonstrate that Thompson sampling enjoys the regret bound

$$\mathrm{Reg}(T) \lesssim d\sqrt{T(\sigma^2 + r^2\mathrm{Tr}(\Sigma_0))\log(1 + T/d)}.$$

In this bound, the *diffusiveness* $\sqrt{\mathrm{Tr}(\Sigma_0)}$ of the prior multiplies the asymptotic minimax rate $d\sqrt{T}$, a looseness that many other regret bounds suffer and that we show is unnecessary.

We separate the minimax terms from a bound on the "burn-in", establishing an $\widetilde{O}(\sigma d\sqrt{T} + dr\sqrt{\mathrm{Tr}(\Sigma_0)})$ regret bound (Section 2), where only the observation noise $\sigma$

scales the minimax rate (and the $\widetilde{O}$ hides terms scaling as $\log(1 + T/d)$). To establish this result, we prove a novel generalization of the elliptical potential lemma (Section 3). This enables a more flexible analysis of stochastic optimization. We interpret the $dr\sqrt{\text{Tr}(\Sigma_0)}$ term as "burn-in" regret that any algorithm necessarily incurrs while reducing the uncertainty captured in the prior distribution to the scale of the observation noise. We establish a lower bound in Section 4 indicating this burn-in regret is unavoidable.

As an additional contribution, in Section 5, we generalize our regret bound to address any strongly log-concave prior and noise distribution, whether or not they are Gaussian.

**Related work** While there is a large body of work on Thompson sampling for the linear bandit, the majority assumes that the coefficients $\theta_\star$ have compact support, and the dependence on the scale of $\theta_\star$ is often opaque. Many papers assume that the model parameter is bounded, while others assume the rewards $R_t$ are bounded, effectively implying the parameter $\theta_\star$ is bounded in the linear bandit (1). We give a somewhat terse list of the most relevant papers, describing briefly the results of their analyses, and recalling that the asymptotic minimax lower bound for regret in the linear bandit (1) scales as $d\sigma\sqrt{T}$ (Rusmevichientong & Tsitsiklis, 2010). We ignore all logarithmic factors below.

- Abeille & Lazaric (2017) study a variant of linear Thompson sampling that inflates the posterior variance by $d$. Assuming that $r = 1$ and $\|\theta_\star\|_2 \leq S$ (Assumption 2), they prove a frequentist (non-Bayesian) regret bound of $\sigma d^{1.5}\sqrt{T} + Sd\sqrt{T}$.

- Agrawal & Goyal (2013) assume that $r = 1$ and $\|\theta_\star\|_2 \leq 1$, and shows a high-probability regret bound of $d^{1.5}\sqrt{T}$. They observe that their regret bound scales linearly with $\|\theta_\star\|_2$ (see Section 2.1 of their paper), which is sub-optimal.

- Dong & Van Roy (2018) assume that the rewards $R_t \in [-1, 1]$, and an inspection of their results to allow $R_t \in [-b, b]$ yields a Bayesian regret bound of $db\sqrt{T}$ (see their Proposition 3, where an application of Pinsker's inequality would result in their "information ratio" $\Gamma_t$ scaling as $db^2$ rather than $d$). Implicitly, then, their regret bounds scale with $\|\theta_\star\|_2$.

- Hamidi & Bayati (2022) assume that $r = 1$ and $\|\theta_\star\|_2 \leq 1$, and an inspection of their proofs suggests regret bounds scale linearly in the potential magnitude of $b = \sup_{a \in \mathcal{A}, \theta_\star \in \Theta} a^\top \theta$. Indeed, inequality (A.2) of the paper becomes

$$A_t^\top \text{Cov}(\theta \mid H_t)A_t \leq b \log(1 + A_t^\top \text{Cov}(\theta \mid H_t)A_t),$$

meaning their regret (implicitly) scales with $\|\theta_\star\|_2$.

- Similar to the preceding results, Russo & Van Roy (2016) assume the rewards belong to an interval of length 1 and the number of actions is finite, proving a Bayesian regret bound of $\sqrt{\log(|\mathcal{A}|)dT}$. Their analysis scales with $b = \sup a^\top \theta$, so while this guarantee can improve upon the typical $d\sqrt{T}$ rate when the action set $\mathcal{A}$ is not too large, it also suffers the same implicit scaling in $\|\theta_\star\|_2$.

- Russo & Van Roy (2013) provide a regret bound that is, in some sense, the closest to ours: they assume rewards $R_t \in [0, C]$, providing a Bayesian regret bound of $\sigma d\sqrt{T} + dC$ for Thompson sampling and an Upper Confidence Bound (UCB)-type algorithm. The only weakness in this result is that they do not allow (potentially) unbounded parameters $\theta_\star$.

- Russo & Van Roy (2014) allow a Gaussian prior (Proposition 5), but assume the action set $\mathcal{A}$ is finite, providing a Bayesian regret bound of $\sigma\sqrt{dT \log |\mathcal{A}|}$. They assume that the prior variance $\Sigma_0$ has diagonal bounded by 1, and they do not explicitly discuss the effect of the prior on the regret.

Kalkanli & Özgür (2020) inspire this particular work, and, as we note in the introduction, they give regret bound of order $\sigma d\sqrt{T} + dr\sqrt{\text{Tr}(\Sigma_0)T}$. We show that the "correct" regret bound—in that there are matching upper and lower bounds—scales as the smaller quantity $\sigma d\sqrt{T} + dr\sqrt{\text{Tr}(\Sigma_0)}$. It is worth mentioning that results on Thompson sampling for Gaussian process bandits (Chowdhury & Gopalan, 2017; Srinivas et al., 2010) also imply regret bounds for the linear-Gaussian bandit. These results face similar limitations as that of Kalkanli & Özgür (2020), where the prior diffusiveness $\sqrt{\|\Sigma_0\|_{\text{op}}}$ multiplies the regret $d\sqrt{T}$.

## 2. Analysis

Our main contribution, which we present in this section, consists of a sharper regret bound for Thompson sampling in the linear-Gaussian bandit (1). To remind the reader, throughout, we wish to bound the Bayesian regret (3), where we assume the canonical Gaussian bandit: we have prior $\theta_\star \sim \mathcal{N}(0, \Sigma_0)$, and assume the actions $\mathcal{A} \subset r\mathbb{B}_2^d$, the $\ell_2$-ball in $\mathbb{R}^d$ of radius $r$. To state the theorem precisely, we require two constants that depend at worst logarithmically on $T$, $r$, $\|\Sigma_0\|_{\text{op}}$, $\sigma^{-1}$, and $1/d$:

$$C_1(d, T) := \sqrt{1 + \max\left\{\frac{24\log T}{d}, \sqrt{\frac{24\log T}{d}}\right\}}$$

$$C_2(d, T, \sigma, r, \Sigma_0) := 2C_1(d, T)\sqrt{\log\left(1 + \frac{r^2\|\Sigma_0\|_{\text{op}}T}{d\sigma^2}\right)}.$$

**Theorem 1.** *Let the preceding assumptions and constant definitions hold. Then Thompson sampling satisfies the Bayesian regret bound*

$$\text{Reg}(T) \leq d\sigma\sqrt{T} \cdot C_2(d, T, \sigma, r, \Sigma_0)$$
$$+ 3\sqrt{2}r\sqrt{d}\text{Tr}(\Sigma_0^{1/2})C_1(d, T) + \sqrt{2r^2\text{Tr}(\Sigma_0)}.$$

Because $C_1$ and $C_2$ only depend logarithmically on $T$, $\sigma^{-1}$, $r$, and $\|\Sigma_0\|_{\text{op}}$, we establish the following corollary:

**Corollary 2.** *Let the assumptions of Theorem 1 hold. Then*

$$\text{Reg}(T) = \widetilde{O}\left(\sigma d\sqrt{T} + dr\sqrt{\text{Tr}(\Sigma_0)}\right).$$

*Proof.* Applying the Cauchy-Schwarz inequality to the trace term in Theorem 1 yields

$$\text{Tr}\left(\Sigma_0^{1/2}\right) \leq \sqrt{\text{Tr}(I_d)\text{Tr}(\Sigma_0)} = \sqrt{d\text{Tr}(\Sigma_0)}. \quad \square$$

Since $\mathbb{E}[\|\theta_\star\|_2^2] = \text{Tr}(\Sigma_0)$, the regret incurred by a very suboptimal action is on the order of $r\sqrt{\text{Tr}(\Sigma_0)}$. Thus, the $dr\sqrt{\text{Tr}(\Sigma_0)}$ term in the regret bound represents the initial exploration cost over all $d$ dimensions. After this burn-in, the concentration of the noise ensures that the posterior localizes to a region predominantly determined by $\sigma$, so that the regret in later rounds is of order $\sigma d\sqrt{T}$, regardless of prior diffusiveness.

## 2.1. Proof of Theorem 1

To simplify notation, define the conditional mean and variance functions $\mathbb{E}_t[\,\cdot\,] := \mathbb{E}[\,\cdot\, \mid H_t]$, $\mathbb{V}_t[\,\cdot\,] := \text{Var}(\cdot \mid H_t)$, and let $V_t := \mathbb{V}_t[\theta_\star]^{-1}$ be the inverse posterior variance (precision matrix). Then (linear-Gaussian) Thompson sampling yields posterior precision matrix

$$V_t = \Sigma_0^{-1} + \frac{1}{\sigma^2}\sum_{i=0}^{t-1} A_i A_i^\top. \quad (4)$$

With this, we may decompose the expected instantaneous regret conditioned on the history $H_t$ via

$$\mathbb{E}_t\left[\theta_\star^\top A_\star - R_{t+1}\right]$$
$$= \mathbb{E}_t\left[\theta_\star^\top (A_\star - A_t)\right]$$
$$= \mathbb{E}_t\left[\theta_\star^\top A_\star - \hat{\theta}_t^\top A_t\right] + \mathbb{E}_t\left[\left(\hat{\theta}_t - \theta_\star\right)^\top A_t\right]$$
$$= \mathbb{E}_t\left[\theta_\star^\top a_\star(\theta_\star) - \hat{\theta}_t^\top a_\star(\hat{\theta}_t)\right] + \mathbb{E}_t\left[\left(\hat{\theta}_t - \theta_\star\right)^\top A_t\right].$$

As $\theta_\star$ and $\hat{\theta}_t$ have the same distribution conditioned on $H_t$, the first term is 0. Thus

$$\mathbb{E}_t\left[\theta_\star^\top A_\star^t - R_{t+1}\right] = \mathbb{E}_t\left[\left(\hat{\theta}_t - \theta_\star\right)^\top A_t\right]. \quad (5)$$

Now, for shorthand, let $\|x\|_B^2 = x^\top Bx$ be the Mahalanobis norm associated to $B$, and for $\beta$ to be determined, define the events

$$E_t(\beta) := \left\{\|\hat{\theta}_t - \theta_\star\|_{V_t} \leq \beta\right\},$$
$$E(\beta) := \bigcap_{t=0}^{T-1} E_t(\beta),$$

which correspond to the sampled $\hat{\theta}_t$ being close (in the appropriate posterior variance metric) to $\theta_\star$. By equation (5), we may decompose the regret into

$$\text{Reg}(T) = \mathbb{E}\underbrace{\left[\mathbf{1}\{E(\beta)\}\sum_{t=0}^{T-1}(\hat{\theta}_t - \theta_\star)^\top A_t\right]}_{(I)}$$
$$+ \mathbb{E}\underbrace{\left[\mathbf{1}\left\{E(\beta)^\complement\right\}\sum_{t=0}^{T-1}(\hat{\theta}_t - \theta_\star)^\top A_t\right]}_{(II)} \quad (6)$$

We control these terms in the remainder of the proof.

### 2.1.1. CONTROLLING TERM (I) OF THE REGRET (6)

Applying the Cauchy-Schwarz inequality gives

$$(I) \leq \mathbb{E}\left[\mathbf{1}\{E(\beta)\}\sum_{t=0}^{T-1}\|\hat{\theta}_t - \theta_\star\|_{V_t}\|A_t\|_{V_t^{-1}}\right]$$
$$\leq \beta\mathbb{E}\left[\sum_{t=0}^{T-1}\|A_t\|_{V_t^{-1}}\right]. \quad (7)$$

To control the sum $\sum_t \|A_t\|_{V_t^{-1}}$, we leverage the particular structure of the precision $V_{t+1}$ as a sum of rank-one updates involving $A_t A_t^\top$, using a new *elliptical potential* lemma. Because the proof of the lemma is involved, we state it here, deferring to Section 3 its proof and commentary on its applications in analysis of bandit algorithms.

**Lemma 3** (Generalized elliptical potential lemma). *Let $V_0$ be a positive definite matrix and*

$$V_{t+1} = V_t + u_t u_t^\top,$$

*where $u_t \in \mathbb{R}^d$ satisfies $\|u_t\|_2 \leq 1$. Then for $p \in [0, 1]$,*

$$\sum_{t=0}^{T-1}\|u_t\|_{V_t^{-1}}^{2p} \leq 2^p T^{1-p}\left(\log\frac{\det V_T}{\det V_0}\right)^p$$
$$+ \frac{3}{2p}\left(\text{Tr}(V_0^{-p}) - \text{Tr}(V_T^{-p})\right),$$

*where for $p = 0$ we take $\lim_{p\downarrow 0}\frac{1}{p}\text{Tr}(V_0^{-p} - V_T^{-p}) = \log\det(V_T/V_0)$.*

By carefully choosing appropriate scaling in inequality (7), we can apply Lemma 3. We thus define

$$U_0 = \frac{\sigma^2}{r^2}V_0 = \frac{\sigma^2}{r^2}\Sigma_0^{-1}, \quad u_t = \frac{A_t}{r},$$
$$U_{t+1} = U_t + u_t u_t^\top = \frac{\sigma^2}{r^2}V_{t+1}. \tag{8}$$

With these choices, $\|u_t\|_{U_t^{-1}} = \sigma^{-1} \cdot \|A_t\|_{V_t^{-1}}$, and taking $p = 1/2$ in Lemma 3, we obtain

$$\frac{1}{\sigma}\sum_{t=0}^{T-1}\|A_t\|_{V_t^{-1}} = \sum_{t=0}^{T-1}\|u_t\|_{U_t^{-1}} \tag{9}$$
$$\leq \sqrt{2T\log\left(\frac{\det(U_T)}{\det(U_0)}\right)} + 3\left(\mathrm{Tr}\left(U_0^{-\frac{1}{2}}\right) - \mathrm{Tr}\left(U_T^{-\frac{1}{2}}\right)\right).$$

We bound each of the terms in the bound (9). Expanding out the log determinant, we obtain

$$\log\frac{\det(U_T)}{\det(U_0)}$$
$$= \log\det\left(U_0^{-1}\left(U_0 + \sum_{t=0}^{T-1}\frac{1}{r^2}A_t A_t^\top\right)\right)$$
$$= \log\det\left(I_d + \frac{1}{\sigma^2}\Sigma_0\sum_{t=0}^{T-1}A_t A_t^\top\right)$$
$$\leq d\log\left(\frac{1}{d}\mathrm{Tr}\left(I_d + \frac{1}{\sigma^2}\Sigma_0\sum_{t=0}^{T-1}A_t A_t^\top\right)\right)$$
$$= d\log\left(1 + \frac{1}{d\sigma^2}\sum_{t=0}^{T-1}A_t^\top\Sigma_0 A_t\right)$$
$$\leq d\log\left(1 + \frac{r^2\|\Sigma_0\|_{\mathrm{op}}}{d\sigma^2}\cdot T\right), \tag{10}$$

where the first inequality uses the arithmetic-geometric inequality and the final bound trivially uses $A_t^\top\Sigma_0 A_t \leq r^2\|\Sigma_0\|_{\mathrm{op}}$. Using definition (8) of $U_0$ and the trivial bound $\mathrm{Tr}(U_0^{-1/2}) - \mathrm{Tr}(U_T^{-1/2}) \leq \frac{r}{\sigma}\mathrm{Tr}(\Sigma_0^{1/2})$, we substitute inequality (10) into inequality (9) to obtain

$$\sum_{t=0}^{T-1}\|u_t\|_{U_t^{-1}}$$
$$\leq \sqrt{2T}\sqrt{d\log\left(1 + \frac{r^2\|\Sigma_0\|_{\mathrm{op}}}{d\sigma^2}\cdot T\right)} + \frac{3r}{\sigma}\mathrm{Tr}\left(\Sigma_0^{\frac{1}{2}}\right).$$

Multiplying through by $\sigma\beta$ gives the term $(I)$ bound

$$(I) \tag{11}$$
$$\leq \beta\sqrt{2T}\sigma\sqrt{d\log\left(1 + \frac{r^2\|\Sigma_0\|_{\mathrm{op}}T}{d\sigma^2}\right)} + 3\beta r\mathrm{Tr}\left(\Sigma_0^{\frac{1}{2}}\right).$$

### 2.1.2. CONTROLLING TERM (II) OF THE REGRET (6)

We use Cauchy-Schwarz to bound the second "small probability" term in equation (6) via

$$(II) \leq \sum_{t=0}^{T-1}\sqrt{\mathbb{E}\left[\mathbf{1}\left\{E(\beta)^{\complement}\right\}^2\right]\mathbb{E}\left[\left((\hat{\theta}_t - \theta_\star)^\top A_t\right)^2\right]}.$$

As posterior sampling makes $\hat{\theta}_t$ and $\theta_\star$ i.i.d. given $H_t$,

$$\mathbb{E}\left[\left((\hat{\theta}_t - \theta_\star)^\top A_t\right)^2\right]$$
$$\leq \mathbb{E}\left[\|\hat{\theta}_t - \theta_\star\|^2\right]r^2 = 2\mathrm{Tr}\left(\mathbb{V}_t\left[\theta_\star\right]\right)r^2 \leq 2\mathrm{Tr}\left(\mathbb{V}\left[\theta_\star\right]\right)r^2.$$

Therefore

$$(II) \leq rT\sqrt{2\mathbb{P}\left(E(\beta)^{\complement}\right)\mathrm{Tr}\left(\Sigma_0\right)}. \tag{12}$$

### 2.1.3. FINALIZING THE PROOF OF THEOREM 1

To finalize the proof, we combine the bounds (11) and (12). For the latter, we choose $\beta$ to make $\mathbb{P}(E(\beta)^{\complement})$ small. Conditioned on $H_t$, $\hat{\theta}_t$ and $\theta_\star$ are i.i.d. Gaussians with covariance $V_t^{-1}$, so $\frac{1}{2}\|\hat{\theta}_t - \theta_\star\|_{V_t}^2$ has $\chi_d^2$-distribution. By standard concentration results (Wainwright, 2019, Ex. 2.11),

$$\mathbb{P}\left[\chi_d^2 - d \geq s\right] \leq \max\left\{e^{-s^2/(8d)}, e^{-s/8}\right\}$$

for $s \geq 0$. Set $s = \max\left\{24\log T, \sqrt{24d\log T}\right\}$. Then $\beta = \sqrt{2d + 2\max\left\{24\log T, \sqrt{24d\log T}\right\}}$ satisfies

$$\mathbb{P}\left[E_t(\beta)^{\complement}\right] \leq \frac{1}{T^3} \quad \text{for all } t \in \{1, \ldots, T\}.$$

By a union bound, we have $\mathbb{P}[E(\beta)^{\complement}] \leq 1/T^2$, whence

$$(II) \leq \sqrt{2\mathrm{Tr}\left(\Sigma_0\right)}r. \tag{13}$$

Plugging inequalities (11) and (13) into equation (6), we obtain

$$\mathrm{Reg}(T) \leq d\sigma\sqrt{T}\sigma C_2(d, T, \sigma, r, \Sigma_0)$$
$$+ 3\sqrt{2}r\sqrt{d}\mathrm{Tr}\left(\Sigma_0^{\frac{1}{2}}\right)C_1(d, T) + \sqrt{2\mathrm{Tr}\left(\Sigma_0\right)}r,$$

where $C_1$ and $C_2$ are as in the theorem statement.

## 3. The generalized elliptical potential lemma

The elliptical potential lemma is a standard tool in the analysis of algorithms for linear bandits (e.g., Dani et al., 2008; Abbasi-Yadkori et al., 2011; Abeille & Lazaric, 2017; Hamidi & Bayati, 2022), often arising in applying a regret decomposition similar to that we provide in inequality (7). The prototypical form relies on controlling quadratic errors, and we restate one version here:

**Lemma 4** (Elliptical potentials, Proposition 2 of Abeille & Lazaric (2017))**.** *Let $\lambda \geq 1$ and $u_0, u_1, \ldots, u_{T-1} \in \mathbb{R}^d$ satisfy $\|u_t\|_2 \leq 1$. Define $V_t = \lambda I + \sum_{s=0}^{t-1} u_s u_s^\top$. Then*

$$\sum_{t=0}^{T-1} \|u_t\|_{V_t^{-1}}^2 \leq 2 \log \frac{\det V_T}{\det V_0}. \qquad (14)$$

While there exist several many generalizations of the elliptical potential lemma, we know of none directly applicable to our setting. In particular, our generalization (Lemma 3) captures the dependence on the initial potential $V_0$, allowing it to be nearly 0, and allows for a more flexible exponent $p$. Before proving Lemma 3, we note a few related works, each of which relies on a related lemma to control the regret in linear bandits or a similar setting:

- Carpentier et al. (2020) control the quantity $\sum_{t=0}^{T-1} \|u_t\|_{V_{t+1}^{-p}}$, while we study $\sum_{t=0}^{T-1} \|u_t\|_{V_t^{-1}}^{2p}$. In addition to the different locations of the $p$ exponent, they rely on $V_0$ being large to show that controlling $\|u_t\|_{V_{t+1}^{-p}}$ is sufficient to bound $\|u_t\|_{V_t^{-p}}$; this difference in indexing precludes more subtle analysis.

- Zhang et al. (2021) generalize the elliptical potential lemma to certain structured monotone convex functions to allow a more sophisticated analysis, but require a number of boundedness assumptions on actions $A_t$ and $\hat\theta_t$.

- To adapt to the norm $\|\theta_\star\|$, Gales et al. (2022) develop adaptive algorithms and analyze their regret by counting the number of times $\|u_t\|_{V_{t-1}^{-1}}$ exceeds a threshold, deriving an elliptical potential count lemma. Because we analyze Thompson sampling, their analyses do not apply here.

In contrast, we generalize the standard result (Lemma 4) to remove the requirement that $V_0 \succeq I_d$ and allow general exponents $p \in [0, 1]$ in the sum $\sum_{t=0}^{T-1} \|u_t\|_{V_t^{-1}}^{2p}$. To do so, we introduce a *burn-in* term that scales like $\mathrm{Tr}(V_0^{-p})$ to capture the initial contribution of small eigenvalues to large values of $\|u_t\|_{V_t^{-1}}^2$. The remainder is captured in the standard $\log(\det V_T / \det V_0)$ term. Before proving the result, we restate the inequality and give commentary. In the context of Lemma 4, we only require that $V_0 \succ 0$, and obtain

$$\sum_{t=0}^{T-1} \|u_t\|_{V_t^{-1}}^{2p} \leq 2^p T^{1-p} \left( \log \frac{\det V_T}{\det V_0} \right)^p$$
$$+ \frac{3}{2p} \left( \mathrm{Tr}(V_0^{-p}) - \mathrm{Tr}(V_T^{-p}) \right). \quad (15)$$

The second term in inequality (15) is tight up to within a factor of $p^{-1}$. For instance, take the standard basis vectors

$u_t = e_t$ for $t \in \{0, \ldots, d-1\}$ and $V_0 = \mathrm{diag}(\lambda_1, \ldots, \lambda_d)$; then

$$\sum_{t=0}^{d-1} \|u_t\|_{V_t^{-1}}^{2p} = \sum_{t=0}^{d-1} \lambda_t^{-p} = \mathrm{Tr}(V_0^{-p}). \qquad (16)$$

The first term in inequality (15) is also tight up to constant factors, even in a regime where the second term is negligible. To see this, assume for simplicity that $T$ is an integer multiple of $d$, take $V_0 = (T/d)I_d$, and define

$$u_t = e^{-1/2} \left( 1 + \frac{d}{eT} \right)^{\lfloor t/d \rfloor / 2} e_{t \bmod d}, \qquad t = 0, \ldots, T-1.$$

Then

$$\|u_t\|_2^2 \leq e^{-1} \left( 1 + \frac{d}{eT} \right)^{T/d} \leq e^{-1} \exp(1/e) < 1.$$

Moreover, each coordinate is updated exactly $T/d$ times, and after the $k$-th update in a given coordinate, the corresponding diagonal entry of $V_t$ equals

$$\frac{T}{d} \left( 1 + \frac{d}{eT} \right)^k.$$

Hence, at every time $t$, $\|u_t\|_{V_t^{-1}}^2 = d/(eT)$ and therefore

$$\sum_{t=0}^{T-1} \|u_t\|_{V_t^{-1}}^{2p} = T \left( \frac{d}{eT} \right)^p = d^p T^{1-p} e^{-p}.$$

On the other hand,

$$\log \frac{\det V_T}{\det V_0} = d \cdot \frac{T}{d} \log \left( 1 + \frac{d}{eT} \right) = T \log \left( 1 + \frac{d}{eT} \right) \leq \frac{d}{e}.$$

Consequently,

$$2^p T^{1-p} \left( \log \frac{\det V_T}{\det V_0} \right)^p \leq 2^p T^{1-p} d^p e^{-p},$$

which matches the left-hand side up to universal constant factors.

Meanwhile, the second term is upper bounded by

$$\frac{3}{2p} \mathrm{Tr}(V_0^{-p}) = \frac{3}{2p} d \left( \frac{d}{T} \right)^p \left( 1 - q^{-pT/d} \right) = d^p T^{1-p} \cdot \frac{3d}{2pT},$$

which is negligible compared to the first term when $T \gg d$. Hence, the first term in (15) is sharp up to universal constant factors.

### 3.1. Proof of Lemma 3

Let $0 < p \leq 1$, as the limiting case $p = 0$ follows trivially. We claim that for all $t$,

$$\|u_t\|_{V_t^{-1}}^{2p} \leq \tag{17}$$

$$\begin{cases} \left(2 \log \frac{\det V_{t+1}}{\det V_t}\right)^p, & \text{if } \|u_t\|_{V_t^{-1}}^2 \leq 2, \\ \frac{3}{2p}(\text{Tr}(V_t^{-p}) - \text{Tr}(V_{t+1}^{-p})), & \text{if } \|u_t\|_{V_t^{-1}}^2 > 2. \end{cases}$$

We first verify the claim (17) for the case when $\|u_t\|_{V_t^{-1}}^2 \leq 2$. By the matrix determinant formula,

$$\det V_{t+1} = \left(1 + u_t^\top V_t^{-1} u_t\right) \det V_t,$$

and rearranging gives

$$\|u_t\|_{V_t^{-1}}^2 = \frac{\det V_{t+1}}{\det V_t} - 1.$$

As $x \leq 2 \log(1 + x)$ for $x \in [0, 2]$, when $\|u_t\|_{V_t^{-1}}^2 \leq 2$,

$$\|u_t\|_{V_t^{-1}}^2 = \frac{\det V_{t+1}}{\det V_t} - 1 \leq 2 \log \frac{\det V_{t+1}}{\det V_t},$$

whence the first claim of inequality (17) follows.

When $\|u_t\|_{V_t^{-1}}^2 > 2$, we use an interpolation argument. By Lewis (1996), as the scalar function $f(x) = -x^p$ is convex for $0 < p \leq 1$, the Hermitian extension $f_{\text{Her}}(M) := -\text{Tr}(M^p)$

(i) is convex on the set of positive definite matrices;

(ii) when $M$ has spectral decomposition $M = U\text{diag}(\lambda(M))U^*$, then its derivative is

$$\nabla f_{\text{Her}}(M) = U\text{diag}\left(\nabla f(\lambda(M))\right)U^*.$$

For $s \in [0, 1]$, define the interpolating function

$$g(s) = -\text{Tr}\left(\left(V_t^{-1} + s\left(V_{t+1}^{-1} - V_t^{-1}\right)\right)^p\right),$$

which satisfies $g(0) = -\text{Tr}(V_t^{-p})$, $g(1) = -\text{Tr}(V_{t+1}^{-p})$, and

$$g'(0) = \left\langle \nabla f_{\text{Her}}(V_t^{-1}), V_{t+1}^{-1} - V_t^{-1}\right\rangle$$
$$= -\text{Tr}\left(pV_t^{-(p-1)}\left(V_{t+1}^{-1} - V_t^{-1}\right)\right).$$

Since $f_{\text{Her}}(M) = -\text{Tr}(M^p)$ is convex in $M$, $g(s)$ is convex in $s$, so $g(1) - g(0) \geq g'(0)$ by first-order convexity. Therefore,

$$\text{Tr}(V_t^{-p}) - \text{Tr}(V_{t+1}^{-p})$$
$$= g(1) - g(0) \geq g'(0) = \text{Tr}\left(pV_t^{1-p}\left(V_t^{-1} - V_{t+1}^{-1}\right)\right).$$

The inversion formula for low rank updates (Sherman-Morrison) implies

$$\text{Tr}(V_t^{-p}) - \text{Tr}(V_{t+1}^{-p}) \geq p\text{Tr}\left(V_t^{1-p} \frac{V_t^{-1} u_t u_t^\top V_t^{-1}}{1 + u_t^\top V_t^{-1} u_t}\right)$$
$$= p \cdot \frac{\|u_t\|_{V_t^{-1-p}}^2}{1 + \|u_t\|_{V_t^{-1}}^2}. \tag{18}$$

With this, we have almost completed the proof, but we must replace the ratio in inequality (18) with one involving $\|u_t\|_{V_t^{-1}}^{2p}$.

We now apply Hölder's inequality to lower bound $\|\cdot\|_{V_t^{-1-p}}$ by $\|\cdot\|_{V_t^{-1}}$ via the following claim.

**Lemma 5.** *Let* $V \succ 0$, $\|u\|_2 \leq 1$ *and* $\|u\|_{V^{-1}}^2 \geq 2$, *and* $0 < p \leq 1$. *Then*

$$\frac{2}{3}\|u\|_{V^{-1}}^{2p} \leq \frac{\|u\|_{V^{-1-p}}^2}{1 + \|u\|_{V^{-1}}^2}.$$

Deferring the proof of Lemma 5 temporarily, we complete the proof of the elliptical potential lemma.

Substituting the result of Lemma 5 into inequality (18) yields

$$\text{Tr}\left(V_t^{-p} - V_{t+1}^{-p}\right) \geq \frac{2p}{3}\|u_t\|_{V_t^{-1}}^{2p}.$$

Combining the cases for $\|u_t\|_{V_t^{-1}}^2 \leq 2$ and $\|u_t\|_{V_t^{-1}}^2 > 2$ in inequality (17), we obtain the unconditional bound

$$\|u_t\|_{V_t^{-1}}^{2p} \leq \left(2 \log \frac{\det V_{t+1}}{\det V_t}\right)^p$$
$$+ \frac{3}{2p}\left(\text{Tr}(V_t^{-p}) - \text{Tr}(V_{t+1}^{-p})\right).$$

Summing over $t \in \{0, \ldots, T - 1\}$ gives

$$\sum_{t=0}^{T-1} \|u_t\|_{V_t^{-1}}^{2p}$$
$$\leq \sum_{t=0}^{T-1} \left(2 \log \frac{\det V_{t+1}}{\det V_t}\right)^p + \frac{3}{2p}\left(\text{Tr}(V_0^{-p}) - \text{Tr}(V_T^{-p})\right)$$
$$\leq 2^p T^{1-p} \left(\log \frac{\det V_T}{\det V_0}\right)^p + \frac{3}{2p}\left(\text{Tr}(V_0^{-p}) - \text{Tr}(V_T^{-p})\right)$$

by Hölder's inequality, completing the proof of Lemma 3.

Finally, we return to the proof of Lemma 5:

*Proof.* Let $V = W\Lambda W^*$ be the eigen-decomposition of $V$, and let $w = W^*u$, so that $\|u\|_{V^q} = \|w\|_\Lambda^q$ for any power $q$.

Then by Hölder's inequality,

$$
\begin{aligned}
\|u\|_{V^{-1}}^2 = \|w\|_{\Lambda^{-1}}^2 &= \sum_{j=1}^d \frac{w_j^2}{\lambda_j} \\
&\leq \left( \sum_{j=1}^d w_j^2 \right)^{\frac{p}{p+1}} \left( \sum_{j=1}^d \frac{w_j^2}{\lambda_j^{1+p}} \right)^{\frac{1}{1+p}} \\
&= \|u\|_2^{\frac{2p}{p+1}} \|u\|_{V^{-1-p}}^{\frac{2}{p+1}},
\end{aligned}
$$

where we use $\|W^* u\|_2 = \|u\|_2$. Because $\|u\|_{V^{-1}}^2 \geq 2$ by assumption, we have $\|u\|_{V^{-1}}^2 \geq \frac{2}{3}(1 + \|u\|_{V^{-1}}^2)$, so

$$
\begin{aligned}
\|u\|_{V^{-1}}^{2p} = \frac{\|u\|_{V^{-1}}^{2p+2}}{\|u\|_{V^{-1}}^2} &\leq \frac{3}{2} \frac{\|u\|_{V^{-1}}^{2p+2}}{1 + \|u\|_{V^{-1}}^2} \leq \frac{3}{2} \frac{\|u\|_2^{2p} \|u\|_{V^{-1-p}}^2}{1 + \|u\|_{V^{-1}}^2}.
\end{aligned}
$$

The result follows from using that $\|u\|_2 \leq 1$. $\qquad\square$

## 4. Tight lower bounds

In this section, we establish a prior-dependent lower bound to show that, generally, any policy must suffer a burn-in term in its regret. We do not quite obtain an instance-specific burn-in that perfectly matches the upper bounds Theorem 1 establishes, but for "non-pathological" priors $\theta_\star \sim \mathcal{N}(0, \Sigma_0)$, we will see it is sharp to within logarithmic factors. To that end, let $\mathfrak{p}$ denote an arbitrary policy, meaning a mapping from histories $H_t$ to distributions $\mathfrak{p}(H_t)$ over actions $A_t$. The Bayesian regret of policy $\mathfrak{p}$ is then

$$
\mathrm{Reg}^{\mathfrak{p}}(T) := \sum_{t=0}^{T-1} \mathbb{E} \left[ \theta_\star^\top A_\star - R_{t+1} \right],
$$

where the expectation integrates over the randomness in the policy $\mathfrak{p}$, the prior, and the noise. We establish the following lower bound, which adapts the lower bound for Gaussian bandits that Rusmevichientong & Tsitsiklis (2010) establish, and whose proof we defer to Appendix A.

**Theorem 6.** *Assume the $d$-dimensional linear-Gaussian bandit (1) with action sets satisfying $r\mathbb{S}^{d-1} \subset \mathcal{A}_t \subset r\mathbb{B}_2^d$ and prior $\theta_\star \sim \mathcal{N}(0, \Sigma_0)$, where $\Sigma_0$ has eigenvalues $\tau_1^2 \geq \cdots \geq \tau_d^2$. Then for $T \in \mathbb{N}$ and any policy $\mathfrak{p}$,*

$$
\mathrm{Reg}^{\mathfrak{p}}(T) \geq \frac{r}{\pi \|\tau\|_2} \sum_{i=2}^d ((i-1) \wedge T) \tau_i^2.
$$

*Additionally, so long as $\Sigma_0 \succ 0$, for $T \in \mathbb{N}$,*

$$
\mathrm{Reg}^{\mathfrak{p}}(T) \geq \sqrt{\frac{2}{\pi}}(d-1)\sigma\sqrt{T} - \frac{\sigma^2}{r} \sqrt{\mathrm{Tr}(\Sigma_0)}\mathrm{Tr}(\Sigma_0^{-1}).
$$

The theorem highlights two regimes: a "burn-in" regime, which depends strongly on the prior $\theta_\star \sim \mathcal{N}(0, \Sigma_0)$, and the long-run regret (though the result holds for all finite samples $T$), which scales as $\sigma d\sqrt{T}$ essentially independently of the prior. Specializations can make the theorem clearer. When the prior covariance $\Sigma_0$ is a scaled multiple of the identity, we obtain the following corollary:

**Corollary 7.** *Let the conditions of Theorem 6 hold and $\Sigma_0 = S^2 I_d$. Then for a numerical constant $c > 0$,*

$$
\mathrm{Reg}^{\mathfrak{p}}(T) \geq c\, S r d^{1/2} \min\{T, d\}.
$$

When the prior covariance $\Sigma_0$ has eigenvalues with the "polynomial" scaling that $\tau_{d-i}^2 = i^{2\alpha}$ for some $\alpha \geq 0$, then $\mathrm{Tr}(\Sigma_0) = \|\tau\|_2^2 \asymp d^{1+2\alpha}$, while $\sum_{i=1}^d (d-i)i^{2\alpha} \asymp d^{2+2\alpha}$, which yields a lower bound matching the upper bounds that Theorem 1 (Corollary 2) provides:

**Corollary 8.** *Let the conditions of Theorem 6 hold, $\Sigma_0$ have polynomially scaling eigenvalues, and $T \geq d$. Then for a numerical constant $c > 0$,*

$$
\mathrm{Reg}^{\mathfrak{p}}(T) \geq c \cdot \frac{r}{\sqrt{\mathrm{Tr}(\Sigma_0)}} d\mathrm{Tr}(\Sigma_0) \gtrsim rd\sqrt{\mathrm{Tr}(\Sigma_0)}.
$$

In each case, Corollary 2 shows Thompson sampling satisfies $\mathrm{Reg}(T) = \widetilde{O}(d\sigma\sqrt{T} + rd\sqrt{\mathrm{Tr}(\Sigma_0)})$, while Theorem 6 shows that for *any* Gaussian prior, $\mathrm{Reg}^{\mathfrak{p}}(T) \gtrsim d\sigma\sqrt{T}$ once $T \geq 2\frac{d\sigma^2}{r^2}\mathrm{Tr}(\frac{1}{d}\Sigma_0)\mathrm{Tr}(\frac{1}{d}\Sigma_0^{-1})^2$. And whenever the number of steps $T \leq r^2\mathrm{Tr}(\Sigma_0)/\sigma^2$, the second burn-in term dominates the lower bound, showing that Thompson sampling is near-optimal in most parameter regimes.

Zhang et al. (2025, Prop. 2, p. 22) provide a minimax lower bound for linear bandits that also exhibits the necessity of a burn-in-type term for deterministic feedback ($\sigma^2 = 0$ in the model (1)). In particular, fixing $\Sigma \succ 0$, $S \geq 0$, and action set $\mathcal{A} = \{a \in \mathbb{R}^d \mid \|a\|_{\Sigma^{-1}} \leq 1\}$, they show for the prior $\theta_\star \sim \mathcal{N}(0, \frac{S^2}{d}\Sigma)$ that

$$
\mathrm{Reg}^{\pi}(T) \geq \frac{S}{\sqrt{d}} \sum_{t=1}^T \left( \mathbb{E}[\|Z\|_2] - \sqrt{t-1} \right)_+,
$$

where $Z \sim \mathcal{N}(0, I_d)$. Because $\mathbb{E}[\|Z\|_2] \geq \sqrt{2d/\pi}$ (see Appendix A), their result implies the lower bound

$$
\mathrm{Reg}^{\pi}(d) \gtrsim Sd^{3/2}
$$

in this case, which matches Corollary 7 with $r = 1$. Their lower bound relies on the behavior of the zero-noise Gaussian setting and the duality relationship between the action set $\mathcal{A}$ and prior variance $\Sigma$, making it so that the results are not always comparable.

## 5. Generalization to strongly-log-concave distributions

While the regret bounds Theorem 1 provides rely on Gaussianity, it is relatively straightforward to extend them to a slightly broader class of distributions whose densities enjoy particular log-concavity properties. To do this, we begin with two definitions:

**Definition 1.** *Let* $\Lambda \succeq 0$. *A function* $f : \mathbb{R}^d \to \mathbb{R}$ *is* $\Lambda$*-strongly convex if* $f(x) - \frac{1}{2}x^\top \Lambda x$ *is convex in* $x$.

**Definition 2.** *A probability distribution* $P$ *with density* $p$ *on* $\mathbb{R}^d$ *is* $\Lambda$*-strongly log-concave if* $x \mapsto -\log p(x)$ *is* $\Lambda$*-strongly convex.*

With these definitions, consider a linear bandit (1) except that instead of noise $W_t \sim \mathcal{N}(0, \sigma^2)$, we assume the rewards

$$R_{t+1} = \theta_\star^\top A_t + W_{t+1}, \quad W_{t+1} \overset{\text{iid}}{\sim} P_W$$

where $P_W$ is strongly $\sigma^{-2}$-strongly-log-concave on $\mathbb{R}$. Let $\theta_\star$ have $\Sigma_0^{-1}$-strongly-log-concave prior and assume as before that the action sets satisfy $\mathcal{A} \subset r\mathbb{B}_2^d$. As in Theorem 1, we define constants

$$C_1(d, T) = \sqrt{1 + \frac{\log T}{d}}$$

$$C_2(d, T, \sigma, r, \Sigma_0) := C_1(d, T)\sqrt{\log\left(1 + \frac{Tr^2\,\|\Sigma_0\|_{\text{op}}}{d\sigma^2}\right)}.$$

Then the following theorem generalizes Theorem 1.

**Theorem 9.** *Let the conditions above hold. Then for a numerical constant* $C < \infty$, *Thompson sampling satisfies*

$$\text{Reg}(T) \leq C\Big[d\sigma\sqrt{T} \cdot C_2(d, T, \sigma, r, \Sigma_0)$$
$$+ r\sqrt{d}\text{Tr}\left(\Sigma_0^{\frac{1}{2}}\right)C_1(d, T) + \sqrt{\text{Tr}\left(\Sigma_0\right)}r\Big].$$

The result follows, *mutatis mutandis*, via the same arguments as those we use to prove Theorem 1, so we defer it to Appendix B.1.

## 6. Conclusions

Analyzing the regret of Thompson sampling in linear-Gaussian bandits, we show that a *burn-in* we may attribute to prior diffusiveness both necessarily appears in regret bounds and decouples additively from the long-run minimax rate $\sigma d\sqrt{T}$. This improves upon existing regret bounds, which scale multiplicatively with prior diffusiveness, and extends to situations in which the noise is log-concave. Limitations, and hence natural areas for extending the approaches here, include (i) the assumptions of log-concavity of the noise

and prior distributions, (ii) the focus on Bayesian regret, and (iii) the (essentially consequent) assumption that the prior is well-specified. Addressing any of these could provide interesting avenues for future work.

## Impact Statement

This is foundational mathematical work. Its primary impact is educational.

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

# A. Proof of Theorem 6

To simplify notation, as in the proof of Theorem 1 define $\mathbb{E}_t[\,\cdot\,] := \mathbb{E}[\,\cdot\,\mid H_t]$ and $\mathbb{V}_t[\,\cdot\,] := \text{Var}(\,\cdot\,\mid H_t)$ to be the conditional expectation and variance. We first adapt Lemma 2.2 of Rusmevichientong & Tsitsiklis (2010) to our setting, which provides the key instantaneous regret bound that we may massage into the two lower bounds in the theorem. In the following lemma, for each $t \in \mathbb{N}$, let $(\boldsymbol{u}_{t,1}, \ldots, \boldsymbol{u}_{t,d})$ be an orthonormal basis of $\mathbb{R}^d$, determined by $H_t$, such that $\boldsymbol{u}_{t,1}$ is parallel to $\mu_t = \mathbb{E}_t[\theta_\star]$, that is, $\boldsymbol{u}_{t,1} = \mu_t / \|\mu_t\|_2$.

**Lemma 10** (Instantaneous Regret Lower Bound). *Let the assumptions of Theorem 6 hold. Then for every policy $\mathfrak{p}$ and $s \leq t \in \mathbb{N}$,*

$$\mathbb{E}_t\left[\theta_\star^\top A_\star - \theta_\star^\top A_s\right] \geq \frac{1}{2} \cdot \mathbb{E}_t\left[\frac{r}{\|\theta_\star\|_2} \sum_{i=2}^{d} \langle \theta_\star - \mathbb{E}_t[\theta_\star], \boldsymbol{u}_{t,i}\rangle^2 + \frac{\|\theta_\star\|_2}{r} \sum_{i=2}^{d} \langle A_s, \boldsymbol{u}_{t,i}\rangle^2\right].$$

*Proof.* As $r\mathbb{S}^{d-1} \subset \mathcal{A}_t \subset r\mathbb{B}_2^d$ for all $t$, the optimal action satisfies $\theta_\star^\top A_\star = r\|\theta_\star\|_2$. Thus for $s \in \mathbb{N}$,

$$\theta_\star^\top A_\star - \theta_\star^\top A_s = r\|\theta_\star\|_2 \left(1 - \frac{\theta_\star^\top}{\|\theta_\star\|_2} \frac{A_s}{\|A_s\|_2}\right) = \frac{1}{2}r\|\theta_\star\|_2 \left\|\frac{\theta_\star}{\|\theta_\star\|_2} - \frac{A_s}{\|A_s\|_2}\right\|_2^2.$$

Decomposing the right hand side in terms of the orthonormal basis $\{\boldsymbol{u}_{t,i}\}_{i=1}^d$, we obtain

$$\begin{aligned}
\theta_\star^\top A_\star - \theta_\star^\top A_s &= \frac{1}{2}r\|\theta_\star\|_2 \sum_{i=1}^{d} \left\langle \frac{\theta_\star}{\|\theta_\star\|_2} - \frac{A_s}{\|A_s\|_2}, \boldsymbol{u}_{t,i}\right\rangle^2 \\
&\geq \frac{1}{2}r\|\theta_\star\|_2 \sum_{i=2}^{d} \left\langle \frac{\theta_\star}{\|\theta_\star\|_2} - \frac{A_s}{\|A_s\|_2}, \boldsymbol{u}_{t,i}\right\rangle^2 \\
&= \frac{1}{2}\sum_{i=2}^{d} \left(\frac{r}{\|\theta_\star\|_2} \langle \theta_\star, \boldsymbol{u}_{t,i}\rangle^2 - 2\langle \theta_\star, \boldsymbol{u}_{t,i}\rangle \langle A_s, \boldsymbol{u}_{t,i}\rangle + \frac{\|\theta_\star\|_2}{r} \langle A_s, \boldsymbol{u}_{t,i}\rangle^2\right).
\end{aligned}$$

Because the sampling distribution of $A_s \sim \mathfrak{p}(H_s)$ enforces that $\mathfrak{p}(H_s)$ is $H_s$-measurable, we observe that $\theta_\star$ and $A_s$ are independent conditional on $H_t$ for any $t \geq s$, and $\{\boldsymbol{u}_{t,i}\}_{i=1}^d$ is $H_t$-measurable. Thus for each $i \in \{2, \ldots, d\}$, $\mu_t = \mathbb{E}_t[\theta_\star]$ is orthogonal to $\boldsymbol{u}_{t,i}$, and so

$$\mathbb{E}_t\left[\langle \theta_\star, \boldsymbol{u}_{t,i}\rangle \langle A_s, \boldsymbol{u}_{t,i}\rangle\right] = \langle \mu_t, \boldsymbol{u}_{t,i}\rangle \mathbb{E}_t\left[\langle A_s, \boldsymbol{u}_{t,i}\rangle\right] = 0.$$

Therefore

$$\mathbb{E}_t\left[\theta_\star^\top A_\star - \theta_\star^\top A_s\right] \geq \frac{1}{2}\mathbb{E}_t\left[\sum_{i=2}^{d} \frac{r}{\|\theta_\star\|_2} \langle \theta_\star, \boldsymbol{u}_{t,i}\rangle^2 + \sum_{i=2}^{d} \frac{\|\theta_\star\|_2}{r} \langle A_s, \boldsymbol{u}_{t,i}\rangle^2\right].$$

The desired result follows once we observe that $\mu_t = \mathbb{E}_t[\theta_\star]$ is orthogonal to $\boldsymbol{u}_{t,2}, \ldots, \boldsymbol{u}_{t,d}$. $\qquad \square$

## A.1. Proof of the first lower bound in Theorem 6

We now prove the first claimed lower bound in the theorem. First, recall Weyl's inequality on the eigenvalues of low rank perturbation of matrices:

**Lemma 11.** *Denote the (real) eigenvalues of a Hermitian matrix $A$ by $\lambda_1(A) \geq \cdots \geq \lambda_d(A)$. Let $A$ be Hermitian and $E$ a Hermitian rank $r$ matrix. Then for $i \in \{1, \ldots, d - r\}$,*

$$\lambda_{i+r}(A + E) \leq \lambda_i(A) \quad \text{and} \quad \lambda_i(A + E) \geq \lambda_{i+r}(A).$$

Intuitively, as each observation only gives a rank-1 update to the posterior covariance matrix, we expect to suffer instantaneous regret scaling as $r$ and a term involving the resolved prior covariance $\Sigma_0$ for at least the first $d$ rounds of the procedure. The next lemma helps to formalize this intuition.

**Lemma 12.** *Let the conditions of Theorem 6 hold and the prior variance $\mathbb{V}_0[\theta_\star] = \Sigma_0$ have eigenvalues $\lambda_1 \geq \cdots \lambda_d \geq 0$. Then for each $t \in \mathbb{N}$,*

$$\mathbb{E}[\theta_\star^\top A_\star - R_{t+1}] \geq \frac{r}{\pi \mathbb{E}[\|\theta_\star\|_2]} \sum_{i=t+2}^{d} \lambda_i,$$

*with the convention that empty sums are zero.*

*Proof.* Without loss of generality we assume $\lambda_d > 0$, as otherwise we simply work in a lower dimensional subspace. Recall the orthogonal decomposition $\{u_{t,i}\}_{i=1}^d$ of $\mathbb{R}^d$ conditional on $H_t$, where $u_{t,1} = \mu_t / \|\mu_t\|_2$ normalizes the expectation $\mu_t = \mathbb{E}_t[\theta_\star]$. We begin by applying Lemma 10 with $s = t$. Since the term $\sum_{i \geq 2} \langle A_t, u_{t,i} \rangle^2$ is nonnegative, we may discard it to obtain

$$\mathbb{E}[\theta_\star^\top A_\star - R_{t+1}] = \mathbb{E}[\theta_\star^\top A_\star - \theta_\star^\top A_t] \geq \frac{r}{2} \mathbb{E}\left[ \frac{1}{\|\theta_\star\|_2} \sum_{i=2}^{d} \langle \theta_\star - \mathbb{E}_t[\theta_\star], u_{t,i} \rangle^2 \right].$$

Multiplying both sides by $\mathbb{E}[\|\theta_\star\|_2]$ and applying the Cauchy-Schwarz inequality (in the form $\mathbb{E}[\sqrt{X}]^2 \leq \mathbb{E}[X/Y]\mathbb{E}[Y]$ for all nonnegative random variables $X, Y$) yields

$$\mathbb{E}[\|\theta_\star\|_2] \, \mathbb{E}[\theta_\star^\top A_\star - R_{t+1}] \geq \frac{r}{2} \cdot \mathbb{E}\left[ \sqrt{\sum_{i=2}^{d} \langle \theta_\star - \mathbb{E}_t[\theta_\star], u_{t,i} \rangle^2} \right]^2.$$

Define the orthogonal random error $M_t := (I - u_{t,1} u_{t,1}^\top)(\theta_\star - \mathbb{E}_t[\theta_\star])$ to satisfy $\|M_t\|_2^2 = \sum_{i=2}^d \langle \theta_\star - \mathbb{E}_t[\theta_\star], u_{t,i} \rangle^2$. Then evidently the expected instantaneous regret at time $t$ has lower bound

$$\mathbb{E}[\theta_\star^\top A_\star - \theta_\star^\top A_t] \geq \frac{r}{2\mathbb{E}[\|\theta_\star\|_2]} \cdot \mathbb{E}[\|M_t\|_2]^2, \tag{19}$$

while conditional on $H_t$, the error has normal distribution

$$M_t \mid H_t \sim \mathcal{N}\left(0, (I - u_{t,1} u_{t,1}^\top)\mathbb{V}_t[\theta_\star](I - u_{t,1} u_{t,1}^\top)\right).$$

We now provide a lower bound on $\mathbb{E}[\|M_t\|_2]$ in inequality (19) using Weyl's inequality. For any positive definite $\Sigma$ and Gaussian vector $Z \sim \mathcal{N}(0, \Sigma)$, letting $|Z|$ denote the elementwise magnitude of $Z$, Jensen's inequality implies

$$\sqrt{\text{Tr}(\Sigma)} \geq \mathbb{E}[\|Z\|_2] = \mathbb{E}[\||Z|\|_2] \geq \|\mathbb{E}[|Z|]\|_2 = \sqrt{\frac{2}{\pi}} \sqrt{\text{Tr}(\Sigma)}.$$

Therefore, recalling the matrix $\boldsymbol{A}_t = [A_0 \ \cdots \ A_{t-1}]^\top \in \mathbb{R}^{t \times d}$ and the cyclic property of the trace, we have

$$\frac{\pi}{2} \cdot \mathbb{E}_t[\|M_t\|_2]^2 \geq \inf_{\|u\|_2 \leq 1} \text{Tr}\left((I - uu^\top)\mathbb{V}_t[\theta_\star](I - uu^\top)\right) = \inf_{\|u\|_2 \leq 1} \text{Tr}\left((I - uu^\top)\left(\Sigma_0^{-1} + \sigma^{-2} \boldsymbol{A}_t^\top \boldsymbol{A}_t\right)^{-1}\right)$$

$$\geq \inf_{\|u\|_2 \leq 1} \inf_{\text{rank}(E) \leq t, E \succeq 0} \text{Tr}\left((I - uu^\top)(\Sigma_0^{-1} + E)^{-1}\right),$$

where we use the prior variance $\Sigma_0 = \mathbb{V}_0[\theta_\star]$. Considering the eigenvalue decomposition $(\Sigma_0^{-1} + E)^{-1} = V\Gamma V^\top$, where the eigenvalues $\gamma_1 \geq \cdots \geq \gamma_d > 0$, we obtain

$$\inf_{\|u\|_2 \leq 1} \text{Tr}\left((I - uu^\top)(\Sigma_0^{-1} + E)^{-1}\right) = \sum_{i=2}^{d} \gamma_i.$$

To relate $\gamma_i$ to the eigenvalues of $\Sigma_0$, let $\mu_1 \leq \cdots \leq \mu_d$ denote the eigenvalues of $\Sigma_0^{-1} + E$. Since $\Sigma_0^{-1}$ has eigenvalues $\lambda_1^{-1} \leq \cdots \leq \lambda_d^{-1}$ and $E \succeq 0$ has rank at most $t$, Weyl's perturbation formula (Lemma 11) implies that

$$\mu_i \leq \lambda_{i+t}^{-1}, \qquad i = 1, \ldots, d - t.$$

Since $\mu_i = \gamma_i^{-1}$, this yields

$$\gamma_i \geq \lambda_{i+t}, \qquad i = 1, \ldots, d - t.$$

Thus $\frac{\pi}{2} \mathbb{E}_t[\|M_t\|_2]^2 \geq \sum_{i=2}^{d-t} \lambda_{i+t}$. Putting the pieces together, we obtain for $t \leq d$ that

$$\mathbb{E}[\|M_t\|_2] \geq \sqrt{\frac{2}{\pi} \inf_{U^\top U = I_{t+1}} \mathrm{Tr}\left((I - UU^\top)\Sigma_0\right)} = \sqrt{\frac{2}{\pi} \sum_{i=t+2}^{d} \lambda_i}.$$

Substitute in inequality (19). $\qquad\square$

We may now finalize the proof of the first lower bound in Theorem 6. By Jensen's inequality, $\mathbb{E}[\|\theta_\star\|_2] \leq \sqrt{\mathrm{Tr}(\Sigma_0)}$, so Lemma 12 implies

$$\mathbb{E}[\theta_\star^\top A_\star - R_{t+1}] \geq \frac{r}{\pi \sqrt{\mathrm{Tr}(\Sigma_0)}} \sum_{i=t+2}^{d} \tau_i^2.$$

Then summing from $t = 0, \ldots, T - 1$, we have

$$\sum_{t=0}^{T-1} \mathbb{E}[\theta_\star^\top A_\star - R_{t+1}] \geq \frac{r}{\pi \|\tau\|_2} \sum_{i=2}^{d} ((i - 1) \wedge T) \tau_i^2.$$

### A.2. Proof of the second lower bound in Theorem 6

We start with the regret lower bound Lemma 10 provides: letting $t = T$ in the lemma and summing yields

$$\sum_{t=1}^{T} \mathbb{E}_T[\theta_\star^\top A_\star - \theta_\star^\top A_t] \geq \frac{1}{2} \sum_{t=1}^{T} \mathbb{E}_T\left[\frac{r}{\|\theta_\star\|_2} \sum_{i=2}^{d} \langle \theta_\star - \mathbb{E}_T[\theta_\star], \boldsymbol{u}_{T,i} \rangle^2 + \frac{\|\theta_\star\|_2}{r} \sum_{i=2}^{d} \langle A_t, \boldsymbol{u}_{T,i} \rangle^2\right].$$

Noting that the first term is independent of the index $t$, taking an outer expectation and using the tower-property, we obtain the regret bound

$$\mathrm{Reg}(T) \geq \frac{1}{2} \cdot \mathbb{E}\left[\sum_{i=2}^{d}\left(\frac{rT}{\|\theta_\star\|_2} \langle \theta_\star - \mathbb{E}_T[\theta_\star], \boldsymbol{u}_{T,i} \rangle^2 + \sum_{t=0}^{T-1} \frac{\|\theta_\star\|_2}{r} \langle A_t, \boldsymbol{u}_{T,i} \rangle^2\right)\right], \tag{20}$$

valid for any policy. Conditional on $H_T$, $\langle \theta_\star - \mathbb{E}_T[\theta_\star], \boldsymbol{u}_{T,i} \rangle$ is a normal random variable with mean zero and variance $\boldsymbol{u}_{T,i}^\top V_T^{-1} \boldsymbol{u}_{T,i}$. Let

$$G_i = \langle \theta_\star - \mathbb{E}_T[\theta_\star], \boldsymbol{u}_{T,i} \rangle / \sqrt{\boldsymbol{u}_{T,i}^\top V_T^{-1} \boldsymbol{u}_{T,i}}.$$

Since $G_i \sim \mathcal{N}(0, 1)$ conditional on $H_T$, it is also standard normal unconditionally. On the other hand, the formula (4) for the posterior precision matrix $V_T$ implies that

$$\sum_{t=0}^{T-1} \langle A_t, \boldsymbol{u}_{T,i} \rangle^2 = \boldsymbol{u}_{T,i}^\top \left(\sum_{t=0}^{T-1} \boldsymbol{A}_t \boldsymbol{A}_t^\top\right) \boldsymbol{u}_{T,i} = \sigma^2 \boldsymbol{u}_{T,i}^\top (V_T - \Sigma_0^{-1}) \boldsymbol{u}_{T,i}.$$

Substituting $G_i$ and this formula into the regret lower bound (20) yields

$$\mathrm{Reg}(T) \geq \frac{1}{2} \cdot \mathbb{E}\left[\sum_{i=2}^{d}\left(\frac{rT}{\|\theta_\star\|_2} G_i^2 \boldsymbol{u}_{T,i}^\top V_T^{-1} \boldsymbol{u}_{T,i} + \frac{\sigma^2 \|\theta_\star\|_2}{r} \boldsymbol{u}_{T,i}^\top V_T \boldsymbol{u}_{T,i}\right)\right] - \mathbb{E}\left[\frac{\sigma^2 \|\theta_\star\|_2}{r} \sum_{i=2}^{d} \boldsymbol{u}_{T,i}^\top \Sigma_0^{-1} \boldsymbol{u}_{T,i}\right].$$

We upper bound the last term and lower bound the first. As $\boldsymbol{u}_{T,1}, \ldots, \boldsymbol{u}_{T,d}$ form an orthonormal basis of $\mathbb{R}^d$,

$$\sum_{i=2}^{d} \boldsymbol{u}_{T,i}^\top \Sigma_0^{-1} \boldsymbol{u}_{T,i} \leq \mathrm{Tr}(\Sigma_0^{-1}).$$

Applying the AM-GM inequality, $ab \geq 2\sqrt{ab}$ for $a, b \geq 0$, to each summand in the first term, we have

$$\frac{rT}{\|\theta_\star\|_2} G_i^2 \boldsymbol{u}_{T,i}^\top V_T^{-1} \boldsymbol{u}_{T,i} + \frac{\sigma^2 \|\theta_\star\|_2}{r} u_{T,i}^\top V_T \boldsymbol{u}_{T,i} \geq 2\sigma\sqrt{T}|G_i|\sqrt{\boldsymbol{u}_{T,i}^\top V_T^{-1} \boldsymbol{u}_{T,i} \cdot \boldsymbol{u}_{T,i}^\top V_T \boldsymbol{u}_{T,i}} \geq 2\sigma\sqrt{T}|G_i|,$$

where the final inequality follows from Cauchy-Schwarz, that is, that $\langle x, y \rangle \leq \sqrt{x^\top B x}\sqrt{y^\top B y}$ for any positive definite $B$. Plugging these back in, we have

$$\text{Reg}(T) \geq \sigma\sqrt{T} \cdot \mathbb{E}\left[\sum_{i=2}^{d} |G_i|\right] - \mathbb{E}\left[\frac{\sigma^2 \|\theta_\star\|_2}{r} \text{Tr}(\Sigma_0^{-1})\right] \geq \sqrt{\frac{2}{\pi}}(d-1)\sigma\sqrt{T} - \frac{\sigma^2}{r}\sqrt{\text{Tr}(\Sigma_0)}\text{Tr}(\Sigma_0^{-1})$$

because $\mathbb{E}[|G_i|] = \sqrt{2/\pi}$ and $\mathbb{E}[\|\theta_\star\|_2] \leq \mathbb{E}[\|\theta_\star\|_2^2]^{1/2} = \sqrt{\text{Tr}(\Sigma_0)}$.

# B. Technical proofs for Section 5

## B.1. Proof of Theorem 9

Our proof relies on concentration properties of random variabels with log-concave distributions. We begin with a standard definition of sub-Gaussian vectors (Vershynin, 2019).

**Definition 3.** *Let $\Sigma$ be positive semidefinite. A random vector $X \in \mathbb{R}^d$ is $\Sigma$-sub-Gaussian if for all $v \in \mathbb{R}^d$*

$$\mathbb{E}\left[\exp\left(v^\top(X - \mathbb{E}[X])\right)\right] \leq \exp\left(\frac{1}{2}v^\top \Sigma v\right).$$

Immediately, we observe that $X \sim \mathcal{N}(\mu, \Sigma)$ is $\Sigma$-sub-Gaussian. Similarly, any random variable with log-concave density is also sub-Gaussian:

**Lemma 13.** *Let $\Lambda$ be positive definite and $X$ have $\Lambda$-strongly log-concave density. Then $X$ is $2\Lambda^{-1}$-sub-Gaussian.*

This lemma follows essentially immediately from Wainwright (2019, Theorem 3.16), but we include a proof for completeness in Section B.1.1. With this, we can fairly straightforwardly demonstrate that any $\Sigma$-sub-Gaussian random vector concentrates, and relatedly, that any random vector with log-concave density similarly concentrates. We have the following technical lemma (again, we defer the proof; see Section B.1.2).

**Lemma 14.** *Let $X$ be a $\Sigma$-sub-Gaussian vector. Then there exists a numerical constant $C < \infty$ such that for all $t \geq 0$,*

$$\mathbb{P}\left(\|X\|_{\Sigma^{-1}} \geq C(\sqrt{d} + t)\right) \leq 2e^{-t^2}.$$

We now turn to the proof of Theorem 9 proper. Recall that $\rho$ denotes the prior distribution of $\theta_\star$, and let $\rho_t = \rho(\cdot \mid H_y)$ denote the posterior density of $\theta_\star$ conditioned on the history $H_t$. Let $p_W$ denote the density of the noise variables $W$. Then by Bayes' rule, for some constant $c$ independent of $\theta$,

$$\log \rho_t(\theta) = c + \log \rho(\theta) + \sum_{s=0}^{t-1} \log p_W(R_{s+1} - A_s^\top \theta),$$

where $c = -\log Z_t$ and $Z_t$ is the normalization constant ensuring that $\rho_t$ integrates to one. Define the inverse variance

$$V_t := \Sigma_0^{-1} + \frac{1}{\sigma^2}\sum_{s=0}^{t-1} A_s A_s^\top.$$

Since $\log p_W(\cdot)$ is $\sigma^{-2}$-strongly concave by assumption, $\log p_W(R_{s+1} - A_s^\top \cdot)$ is $A_s A_s^\top/\sigma^2$-strongly concave, and so the posterior $\rho_t$ on $\theta_\star$ is $V_t$-strongly log-concave.

As in the proof of Theorem 1, for a constant $\beta$ to be determined later, define the events

$$E_t(\beta) := \left\{\|\hat\theta_t - \theta_\star\|_{V_t} \leq \beta\right\},$$

$$E(\beta) := \bigcap_{t=0}^{T-1} E_t(\beta).$$

We use the regret decomposition (6),

$$\text{Reg}(T) = \underbrace{\mathbb{E}\left[\mathbf{1}\{E(\beta)\}\sum_{t=0}^{T-1}(\hat{\theta}_t - \theta_\star)^\top A_t\right]}_{(I)} + \underbrace{\mathbb{E}\left[\mathbf{1}\left\{E(\beta)^{\complement}\right\}\sum_{t=0}^{T-1}(\hat{\theta}_t - \theta_\star)^\top A_t\right]}_{(II)}.$$

Then inequalities (11) and (12), which rely on no probabilistic structure of the iterates, imply

$$(I) \le \beta\sqrt{2T}\sigma\sqrt{d\log\left(1 + \frac{r^2\|\Sigma_0\|_{\text{op}}T}{d\sigma^2}\right)} + 3\beta r\text{Tr}\left(\Sigma_0^{\frac{1}{2}}\right) \;\text{ and }\; (II) \le rT\sqrt{2\mathbb{P}(E(\beta)^{\complement})\text{Tr}\left(\Sigma_0\right)}.$$

We now simply choose a suitable $\beta$ for this log-concave setting. Since $\log\rho_t(\cdot)$ is $V_t$-strongly concave, by Lemma 13, $\theta_\star \mid H_t$ is $2V_t^{-1}$-sub-Gaussian (that is, conditional on the history $H_t$). So $\hat{\theta}_t - \theta_\star \mid H_t$ is $4V_t^{-1}$-sub-Gaussian. Thus, taking $t = \sqrt{3\log T}$ in Lemma 14, we see that by taking

$$\beta = O(1)\sqrt{d + \log T},$$

then a union bound implies for all $t \in \{0, \ldots, T-1\}$,

$$\mathbb{P}\left[E(\beta)^{\complement}\right] \le \frac{2}{T^2}.$$

Combining the pieces gives the theorem.

### B.1.1. PROOF OF LEMMA 13

As we note, the proof is a more or less trivial adaptation of the argument to prove Theorem 3.16 of Wainwright (2019). First, we recall the Prékopa-Leinder inequality (e.g. Ball, 1997), which states that if $u, v, w$ are non-negative integrable functions satisfying the log-concavity-type inequality

$$w(\lambda x + (1 - \lambda)y) \ge u(x)^\lambda v(y)^{1-\lambda}, \quad \text{all } x, y \tag{21}$$

for some $\lambda \in [0, 1]$, then

$$\int w(x)dx \ge \left(\int u(x)dx\right)^\lambda \left(\int v(x)dx\right)^{1-\lambda}. \tag{22}$$

Moving to the proof proper, assume without loss of generality that $\mathbb{E}[X] = 0$, since we can always shift the distribution by $-\mathbb{E}[X]$. Fix $v \in \mathbb{R}^d$, and define the infimal convolution of $x \mapsto v^\top x$ and $\frac{1}{4}\|\cdot\|_\Lambda^2$ by

$$g(y) := \inf_x\left\{v^\top x + \frac{1}{4}(x-y)^\top\Lambda(x-y)\right\} = v^\top y - v^\top\Lambda^{-1}v.$$

Define $\psi(x) := -\log p(x)$, which is $\Lambda$-strongly convex by assumption. To apply inequality (22), the auxiliary functions

$$w(z) := p(z) = \exp(-\psi(z)), \quad u(x) := \exp(-v^\top x - \psi(x)), \quad v(y) := \exp(g(y) - \psi(y)).$$

We verify that inequality (21) holds for $\lambda = \frac{1}{2}$ with these choices:

$$\frac{1}{2}\log u(x) + \frac{1}{2}\log v(y) - \log w\left(\frac{1}{2}x + \frac{1}{2}y\right)$$

$$= \left(-\frac{1}{2}v^\top x - \frac{1}{2}\psi(x)\right) + \left(\frac{1}{2}g(y) - \frac{1}{2}\psi(y)\right) + \psi\left(\frac{1}{2}x + \frac{1}{2}y\right)$$

$$= \frac{1}{2}\left(g(y) - v^\top x - \frac{1}{4}(x-y)^\top\Lambda(x-y)\right) + \left(\psi\left(\frac{x+y}{2}\right) + \frac{1}{8}(x-y)^\top\Lambda(x-y) - \frac{1}{2}\psi(x) - \frac{1}{2}\psi(y)\right).$$

The first term is non-positive by the definition of $g(y)$, and the second term is non-positive by the $\Lambda$-strong convexity of $\psi(x)$. Thus,

$$\frac{1}{2}\log u(x) + \frac{1}{2}\log v(y) \leq \log w\left(\frac{1}{2}x + \frac{1}{2}y\right).$$

We may therefore apply inequality (22) to obtain

$$1 = \int_{\mathbb{R}^d} w(x)dx \geq \left(\int_{\mathbb{R}^d} u(x)dx\right)^{\frac{1}{2}} \left(\int_{\mathbb{R}^d} v(x)dx\right)^{\frac{1}{2}} = \mathbb{E}\left[e^{-v^\top X}\right]^{\frac{1}{2}} \mathbb{E}\left[e^{g(X)}\right]^{\frac{1}{2}}.$$

Rearranging, we obtain

$$\mathbb{E}\left[e^{g(X)}\right] \leq \mathbb{E}\left[\exp\left(-v^\top X\right)\right]^{-1} \leq \exp\left(-v^\top \mathbb{E}[X]\right)^{-1} = 1$$

by Jensen's inequality. Plugging in the formula for $g(y)$ gives

$$\mathbb{E}\left[\exp\left(v^\top X\right)\right] \leq \exp\left(v^\top \Lambda^{-1} v\right).$$

As $v \in \mathbb{R}^d$ was arbitrary, $X$ is $2\Lambda^{-1}$-sub-Gaussian.

### B.1.2. PROOF OF LEMMA 14

Define $\psi_2(x) = e^{x^2} - 1$. Then the *Orlicz-$\psi_2$-norm* (e.g. Wainwright, 2019, Ch. 5.6) of a random variable $Y$ is

$$\|Y\|_{\psi_2} := \inf\left\{t > 0 \mid \mathbb{E}[e^{Y^2/t^2}] \leq 2\right\}.$$

Following Wainwright (2019), we say stochastic process vector $\{Y_u\}_{u \in \mathbb{R}^d}$ is a *b-Orlicz-$\psi_2$-process* if

$$\|Y_\theta - Y_{\theta'}\|_{\psi_2} \leq b\|\theta - \theta'\|_2.$$

For a compact set $K \subset \mathbb{R}^d$ and metric $\rho$ on $K$, let $\text{diam}_\rho(K) = \sup_{u,v \in K} \rho(u,v)$ be the $\rho$-diameter of $K$, and let $N(\epsilon, K, \rho)$ denote the $\rho$-covering number of $K$ at radius $\epsilon$. Define the entropy integral

$$J(K, \rho) := \int_0^{\text{diam}_\rho(K)} \sqrt{\log(1 + N(\epsilon, K, \rho))}d\epsilon$$

(where we recall that $\psi_2^{-1}(z) = \sqrt{\log(1+z)}$). Then (Wainwright, 2019, Thm. 5.36) there exists a finite numerical constant $C < \infty$ such that for any $t \geq 0$ and any compact $K$, any $b$-Orlicz process satisfies

$$\mathbb{P}\left(\sup_{u,v \in K}|Y_u - Y_v| \geq C(J(K, b\|\cdot\|_2) + t)\right) \leq 2\exp\left(-\frac{1}{b^2}\frac{t^2}{\text{diam}_2^2(K)}\right), \tag{23}$$

where $\text{diam}_2$ denotes the $\ell_2$-diameter.

We now demonstrate that $Y_u := u^\top \Sigma^{-1/2} X$ is an $O(1)$-Orlicz process. Indeed, for any $u, v$, we have

$$\mathbb{E}\left[\exp\left((v-u)^\top \Sigma^{-1/2} X\right)\right] \leq \exp\left(\frac{1}{2}\|v-u\|_2^2\right),$$

so that $Y_u := u^\top \Sigma^{1/2} X$ is an $O(1)$-Orlicz-$\psi_2$-process by any of the equivalent definitions of sub-Gaussianity (Buldygin & Kozachenko, 2000). On the ball $\mathbb{B}_2^d$, the covering number $N(\epsilon, \mathbb{B}_2^d, \|\cdot\|_2) \leq (1 + \frac{2}{\epsilon})^d$, yielding entropy integral

$$J(\mathbb{B}_2^d, \|\cdot\|_2) \lesssim \int_0^2 \sqrt{d\log\left(1 + \frac{2}{\epsilon}\right)}d\epsilon \lesssim \sqrt{d}.$$

Set $b = 1$ in inequality (23) and take $K = \mathbb{B}_2^d$.

