# OpenReview forum: "Prior Diffusiveness and Regret in the Linear-Gaussian Bandit"
_ICML.cc/2026/Conference — ICML 2026 regular_

### Official Review · Reviewer_97yx · 2026-02-24

**Soundness:** 3
**Presentation:** 3
**Significance:** 2
**Originality:** 3
**Overall Recommendation:** 4
**Confidence:** 4

**Summary:**

The submission proposes a new Bayesian analysis of linear Thompson Sampling under Gaussian prior and noise, manifesting that the "typical magnitude" of the ground-truth parameter can be decoupled from the $\sqrt{T}$-type term in the Bayesian regret bound when there are no explicit assumptions on the boundedness of rewards or that the parameter space is compact. The authors also extends their analysis to tackle strongly log-concave priors (and noises). At the heart of the analysis is a new elliptical potential lemma that holds even if $\lambda_\max(\Sigma_0)$ is sufficiently large (at cost of an additional burn-in term) and consequently bypasses the usual $\min\\{1, \cdot\\}$-type truncation in the frequentist analysis. The authors also demonstrate the necessity of the burn-in term on special cases where $\Sigma_0$ admits a power-law spectrum.

**Compliance With Llm Reviewing Policy:**

Affirmed.

**Final Justification:**

The authors answered my question on the burn-in cost affirmatively. I keep my positive evaluation.

**Key Questions For Authors:**

NA

**Limitations:**

Yes.

**Strengths And Weaknesses:**

### Strengths

1. The elliptical potential lemma established in the paper is new.
2. The submission bypasses the usual $\min\{1, \cdot\}$-type truncation in the analysis for linear bandits, which enables the decoupling of the magnitudes of rewards and ground-truth parameter from $\mathrm{poly}(T)$ in the regret bound.
3. The proof of Lemma 3 is well-written and easy to follow.

### Weaknesses

1. Actually, boundedness of rewards and ground-truth parameter is not a strong assumption in the frequentist setting. The authors are encouraged to provide more apple-to-apple discussions on the assumptions in the previous works, especially those analyzing the Bayesian regret.
2. Minor question: does the lower bound in this submission imply that LinUCB will also has a burn-in term in the Bayesian setting?
3. The frameworks for the proof of upper and lower bounds both do not deviate much from the standard ones.

---

> ### Author Rebuttal · Authors · 2026-03-27
>
> We thank the reviewer for the constructive review and clear feedback.
>
> > Minor question: does the lower bound in this submission imply that LinUCB will also has a burn-in term in the Bayesian setting?
>
> Indeed, this lower bound in the Bayesian setting applies to all algorithms. In addition, in response to questions raised by reviewer bomQ, we have strengthened our Theorem 6 to include the following long-run lower bound on regret for arbitrary priors:
>
>   Assume the $d$-dimensional linear-Gaussian bandit with action sets satisfying $r  \mathbb{S}^{d-1} \subset \mathcal{A}_t \subset r \mathbb{B}_2^d$ and prior
>   $\theta\_\* \sim \mathcal{N}(0, \Sigma_0)$.
>   Then for $T \in \mathbb{N}$ and any policy $\mathfrak{p}$, so long as $\Sigma_0 \succ 0$, for $T \in \mathbb{N}$,
>   \begin{align*}
>     \mathrm{Reg}^\mathfrak{p}(T)
>     \ge \sqrt{\frac{2}{\pi}} (d - 1) \sigma \sqrt{T}
>     - \frac{\sigma^2}{r} \sqrt{\mathrm{Tr}(\Sigma\_0)}
>     \mathrm{Tr}(\Sigma\_0^{-1}).
>   \end{align*}
>
> In particular, this addition characterizes the
> long-run regret (though the result holds for all finite samples $T$), which
> scales as $\sigma d \sqrt{T}$ essentially independently of the prior.

---

> > ### Author Rebuttal · Reviewer_97yx · 2026-03-31
> >
> > The authors answered my question on the burn-in cost affirmatively.

---

### Official Review · Reviewer_vmdR · 2026-03-09

**Soundness:** 3
**Presentation:** 4
**Significance:** 3
**Originality:** 3
**Overall Recommendation:** 5
**Confidence:** 4

**Summary:**

This paper studies the Bayesian regret in Linear-Gaussian bandits. It devises a new elliptical potential lemma, which helps to improve the dependence on the prior distribution from the multiplicative dependence to the additive dependence. A tight lower bound is established to show that the dependence on this prior is tight.

**Compliance With Llm Reviewing Policy:**

Affirmed.

**Final Justification:**

The paper improves the elliptical lemma used in  linear bandits, which is fundamental. Hence, it should be accepted.

**Key Questions For Authors:**

- I would appreciate it if the authors can list a few applications of the established results in the linear bandits literature. This would further demonstrate the significance of the work.

**Limitations:**

yes

**Strengths And Weaknesses:**

**Strengths**:
- The paper is clearly written and concise. The comparison with the existing linear bandits literature is sound.
- The new elliptical potential lemma is novel and fundamental in the linear bandit literature. I expect this new result can improve the results in many other derivative works in linear bandit/RL problems.
- The generalized elliptical potential lemma is both elegant and practical, providing a flexible tool for future analyses in related domains.

**Weaknesses**:
- While I appreciate the technical contributions, it would be better if the authors move the proofs to the appendix and present more results/extensions in the paper. For instance, how the improved results can further enhance the results in sparse linear bandits, etc. This helps to enrich the content of the manuscript.
- As the authors confessed, current techniques apply to the Bayesian regret only. An extension to the Frequentist regret can be more interesting. In addition, the log-concave assumption on the noise and prior might be restrictive.

minors:
- Line 294 right: I suppose the power of $V$ in the numerator should be $-1-p$.

---

> ### Author Rebuttal · Authors · 2026-03-27
>
> We thank the reviewer for the constructive review and helpful feedback. We have fixed the typo on line 294.
>
> We agree that explicitly demonstrating downstream applications would further strengthen the paper. Our primary goal here is to develop the generalized elliptical potential lemma as a foundational technical tool and use it to obtain sharper Bayesian regret bounds in the linear-Gaussian setting. While we do not include additional applications in the current version,
> the lemma directly generalizes the classical elliptical potential lemma, and thus can be used as a drop-in replacement in analyses that rely on it, potentially leading to sharper bounds in those settings.

---

> > ### Author Rebuttal · Reviewer_vmdR · 2026-04-01
> >
> > Thank the authors for their response. I do not have any other questions.

---

### Official Review · Reviewer_bomQ · 2026-03-12

**Soundness:** 4
**Presentation:** 3
**Significance:** 4
**Originality:** 4
**Overall Recommendation:** 6
**Confidence:** 4

**Summary:**

There are two main contributions of this paper. First, a tighter Bayesian regret upper bound for Thompson sampling in Linear-Gaussian bandits is proven, which decouples the prior-dependent "burn-in" term from the leading $\sqrt{T}$ term. The key technical tool for this is a novel generalized elliptical potential lemma (Lemma 3), which may be of independent interest. Second, a lower bound is proven that shows that such "burn-in" is unavoidable.

**Compliance With Llm Reviewing Policy:**

Affirmed.

**Final Justification:**

I have no further concerns. I have updated my score.

**Key Questions For Authors:**

1. The authors briefly mention that the second term in the generalized elliptical lemma is tight up to a factor of $p^{-1}$. How about the first term? Is that also tight? For the fun of it, I scratched some stuff below, but maybe the authors could make it tighter.
     - I wonder if this can be utilized to derive a tighter (Bayesian) regret lower bound up to logarithmic factors.

2. Throughout, the authors seem to ignore the $-\mathrm{Tr}(V_T^{-p})$ term at the end by simply bounding it above with $0$. Are there any cases (either in the context of this paper or even beyond) where keeping the term may lead to something tighter?

3. Is it common in the Thompson sampling + linear-Gaussian bandits literature to consider polynomially scaling eigenvalues structure for $\Sigma_0$? If yes, then is there really no known Bayesian regret lower bound in this scenario as well, or could we somehow adopt the proof of Rusmevichientong & Tsitsiklis (2010, Theorem 2.1)? This would make the paper even stronger.


### Tightness of the first term
I largely follow the argument of Li et al. (2019, Lemma 11) for $d = 1$ and $V_0 = 1$, but maybe this could be extended to $d > 1$ as well.
For reason to be clear later, we will only deal with the case when $p \geq \frac{1}{2}$.

Define $S_t = \left( 1 + \frac{(1 - p) \log T}{T} \right)^t$ for $t \geq 0$ and $z_t = \sqrt{\frac{p S_{t-1} \log T}{T} }$ for $t \geq 1$. Then $z_t$ is monotone increasing in $t$ and $S_{T-1} \leq T^{1 - p}$. Then, we have that $z_t \in [0, 1]$ since
$$z_t \leq z_T = \sqrt{ \frac{p S_{T-1} \log T}{T} } \leq \sqrt{ \frac{p \log T}{T^p} } \leq 1$$

Now, note that
$$
V_t = 1 + \sum_{j=1}^t z_j^2 = 1 + \sum_{j=1}^t \frac{p \left( 1 + \frac{(1 - p) \log T}{T} \right)^{j-1} \log T}{T}
= 1 + \frac{p \log T}{T} \frac{\left( 1 + \frac{(1 - p) \log T}{T} \right)^t - 1}{\left( 1 + \frac{(1 - p) \log T}{T} \right) - 1}
= 1 + \frac{p}{1 - p} \left( S_t - 1 \right)
\leq \frac{p}{1 - p} S_t,
$$
where the inequality is because $p \geq \frac{1}{2}$.

Therefore, we have
$$
\sum_{t=1}^T (z_t^2 / V_{t-1})^p \geq \sum_{t=1}^T \left( \frac{p S_{t-1} \log T}{T} \frac{1 - p}{p S_{t-1}} \right)^{p} = (1 - p)^p T^{1-p} (\log T)^{1-p}
$$

This shows that in terms of $T$, $T^{1-p} (\log T)^p$ is tight when $p \geq \frac{1}{2}$. When $p < \frac{1}{2}$, I have no idea ;)

https://ieeexplore.ieee.org/document/10103667


=====
(update, 2026.04.01) I have raised my score from 5 to 6 after reading through the author's response.

**Limitations:**

yes

**Strengths And Weaknesses:**

**Strengths**
- Compact and very well-written. I truly enjoyed reading this work.
- Discussions and relations to prior works are very well-done
- Simple yet elegant proof that leads to minimax optimal Bayesian regret for linear-Gaussian bandits.


**Weaknesses**
The only weakness of the paper is that some parts of the writing should be clarified, and there are some typos to be fixed:
- In the proof of Lemma 5, $\lVert w \rVert_{V^{-1}}^2$ should be $\lVert w \rVert_{\Lambda^{-1}}^2$.

- In lines 355-356, it should be Zhang et al. (2025, Prop. 2), not Prop. 2.1. Also, please consider putting in the precise page number (I think it is pg. 22 in the PMLR version), as the provided bound is part of the proof of the Proposition, not the Proposition itself.
 - Below Definition 2, "$P_W$ is strongly $\sigma^{-2}$-strongly-log-concave" => "$P_W$ is strongly $\sigma^{-2} I$-strongly-log-concave"

- For the sake of clarity in Lemma 12, please explicitly state the convention for the empty sum, namely that $\sum_{i=t+1}^d \lambda_i = 0$ when $t > d$.

- In the chain of inequalities in the beginning of the proof of Lemma 12, please be kinder in which result implies with. To my understanding, Lemma 10 implies the first inequality; Cauchy-Schwarz implies the second inequality; Eqn. (19) is then implied by considering $\mathbb{E}[\lVert \theta_\star \rVert_2] \cdot \mathbb{E}[\theta_\star^\top A_\star - \theta_\star^\top A_t]$ and the last part of the chain of inequalities. Personally, I find using \tag{} very helpful.

- In lines 515-517, I don't think $\sqrt{\mathrm{Tr}(\Sigma)} \geq \mathbb{E}[\lVert Z \rVert_2]$ is necessary.

- In the proof of Lemma 12, please be a bit more kind in explaining how Weyl's perturbation formula implies $\gamma_i \geq \lambda_{i+t}$. Due to inverses being involved, I don't think this is like too trivial.

- In lines 533-535, I believe the sum in the square root should be from $i = t+2$ to $d$, not $i = t+1$. Also, the same typo for Lemma 12.

- In lines 541-543, I don't understand why we suddenly require the variational form involving $U \in St(d, t+1)$. isn't 546-549 directly implied by Lemma 12 without having to go through such form?

- Below Lemma 14, for clarity recall that $\rho$ is the prior of $\theta_\star$. Also, what is this constant $c$?

- The discussions below Corollary 8 are slightly imprecise in that the authors should separate the optimality claim according to the choice of $\Sigma_0$. To my understanding,
   - Let us choose $S = d^{-1/2}$ and consider the case when $\Sigma_0 = S^2 I_d = (1 / d) I_d$. Then, Rusmevichientong & Tsitsiklis (2010, Theorem 2.1) shows a Bayesian regret lower bound of $\Omega(d \sigma \sqrt{T})$. Thus, combining with the paper's Corollary 7, when $T \geq d$, the final regret bound takes the form of $\Omega(d \sigma \sqrt{T} + r d)$. This *precisely* matches the regret upper bound (up to logarithmic factors), and thus here, I agree that the result is optimal.
   - However, when $\Sigma_0$ is not of scaled identity, then to my understanding, there is no known $d \sigma \sqrt{T}$ lower bound. Thus, here, one can only claim optimality for the burn-in term.

---

> ### Author Rebuttal · Authors · 2026-03-26
>
> We thank the reviewer for the constructive review and clear feedback. We have fixed the math and style issues accordingly.
>
> > In lines 355-356, it should be Zhang et al. (2025, Prop. 2), not Prop. 2.1. Also, please consider putting in the precise page number (I think it is pg. 22 in the PMLR version), as the provided bound is part of the proof of the Proposition, not the Proposition itself.
>
> Thank you for pointing this out. We were mixing the proposition numbers in their arxiv version and the PMLR version. We will add the precise page number as recommended.
>
> > Below Lemma 14, for clarity recall that $\rho$ is the prior of $\theta_*$. Also, what is this constant $c$?
>
> The constant $c$ is the (log) normalizing constant arising from Bayes’ rule, i.e., $c = -\log Z_t$, where $Z_t$ ensures that the posterior density integrates to one. Importantly, $c$ does not depend on $\theta$. We have replaced "chain rule" by "Bayes' rule" for clarity, and update the paper accordingly.
>
> # Tightness of general elliptical potential lemma
>
> Thank you for raising this.
>
> The first term is also tight up to constant factors, even in a regime where the second term is negligible.
> To see this, assume for simplicity that $T$ is an integer multiple of $d$, take
> $V_0 = (T/d) I_d,$
> and define
> $$u_t=
> e^{-1/2} \left(1+\frac{d}{eT}\right)^{\lfloor t/d \rfloor/2} e_{t \bmod d},\qquad t=0,\dots,T-1.$$
> Then
> $$
> \|u_t\|_2^2
> \le
> e^{-1} \left(1+\frac{d}{eT}\right)^{T/d}
> \le
> e^{-1}\exp\left(1/e\right)
> < 1.
> $$
>
> Moreover, each coordinate is updated exactly $T/d$ times, and after the
> $k$-th update in a given coordinate, the corresponding diagonal entry of
> $V_t$ equals
> $$\frac{T}{d} \left(1 + \frac{d}{eT}\right)^k.$$
> Hence, at every time $t$,
> $$ \|u\_t\|\_{V_t^{-1}}^2=\frac{d}{eT}, $$
> and therefore
> $$\sum_{t=0}^{T-1} \|u_t\|_{V_t^{-1}}^{2p}=T \left(\frac{d}{eT}\right)^p=d^p T^{1-p} e^{-p}.$$
>
> On the other hand,
> $$\log \frac{\det V_T}{\det V_0}=d \cdot \frac{T}{d} \log\left(1+\frac{d}{eT}\right)=T \log\left(1+\frac{d}{eT}\right)\le\frac{d}{e}.$$
> Consequently,
> $$2^p T^{1-p}\left(\log \frac{\det V_T}{\det V_0}\right)^p\le2^p T^{1-p} d^p e^{-p},$$
> which matches the left-hand side up to universal constant factors.
>
> Meanwhile, the second term is upper bounded by
> $$\frac{3}{2p} \mathrm{Tr}(V_0^{-p})=\frac{3}{2p} d \left(\frac{d}{T}\right)^p,$$
> which is smaller than $d^p T^{1-p}$ by a factor of $d/T$.
> Hence, when $T \gg d$, the second term is negligible, and the first term
>  is sharp up to universal constant
> factors.
>
> Thus, both terms in the generalized elliptical potential lemma are tight (up to constants) in complementary regimes.
>
> We also added the above discussion to the paper.
>
> # The trace term
>
> The term $\mathrm{Tr}(V_T^{-p})$ decays quickly with $T$ and doesn't matter in any case we are aware of.
>
> # Lower bound for arbitrary priors
>
> Thank you for raising this point, as it inspired us to revisit the lower bound and strengthen in. While we cannot upload the full proof here (as we cannot add extra files, and it would overflow the character limits), we have adapted the techniques of Rusmevichientong & Tsitsiklis (2010, Theorem 2.1), to strengthen our Theorem 6 to also show that the long-run regret has a lower bound that is essentially independent of the prior and of order $\sigma d \sqrt{T}$:
>
>   Assume the $d$-dimensional linear-Gaussian bandit with action sets satisfying $r  \mathbb{S}^{d-1} \subset \mathcal{A}_t \subset r \mathbb{B}_2^d$ and prior
>   $\theta\_\* \sim \mathcal{N}(0, \Sigma_0)$.
>   Then for $T \in \mathbb{N}$ and any policy $\mathfrak{p}$, so long as $\Sigma_0 \succ 0$, for $T \in \mathbb{N}$,
>   \begin{align*}
>     \mathrm{Reg}^\mathfrak{p}(T)
>     \ge \sqrt{\frac{2}{\pi}} (d - 1) \sigma \sqrt{T}
>     - \frac{\sigma^2}{r} \sqrt{\mathrm{Tr}(\Sigma\_0)}
>     \mathrm{Tr}(\Sigma\_0^{-1}).
>   \end{align*}
> We are glad the reviewer raised the question!

---

> > ### Author Rebuttal · Reviewer_bomQ · 2026-04-01
> >
> > I thank the authors for the comprehensive response. All my concerns and questions have been resolved, and I'm raising my score. I believe this is a very neat work that closes the known gap in Bayesian linear bandits via a clever technical lemma (Lemma 3) that would be of interest to the bandit community.
> > I look forward to the revision incorporating all feedback from myself and other reviewers, especially the strengthened, prior-agnostic lower bound.

---

### Official Review · Reviewer_zKca · 2026-03-19

**Soundness:** 3
**Presentation:** 3
**Significance:** 3
**Originality:** 3
**Overall Recommendation:** 5
**Confidence:** 3

**Summary:**

This paper analyzes the Bayesian Regret of Thompson Sampling for linear bandits under the assumption that the action set is bounded, but the prior is Gaussian with covariance $\Sigma_0$. Contrary to previous works that either assume a compact prior or have a term in the regret scaling with $d\sqrt{T \text{Tr}(\Sigma_0)}$, they improved the scaling of this term to $d\sqrt{\text{Tr}(\Sigma_0)}$, effectively removing the time horizon dependence in this term. This improvement makes the regret upper bound of this Bayesian algorithm more in line with those of other state-of-the-art algorithms for linear bandits in terms of dependence on the norm of the true parameter. To do so, they developed a novel result on the elliptic potential lemma, which is one of the main messages of this paper and could be of separate interest. They also provide a lower bound of the Bayesian regret, matching (constant and polylog term omitted) their upper bound.

**Compliance With Llm Reviewing Policy:**

Affirmed.

**Final Justification:**

I think the presentation is clear and the paper technically solid; I only had minor comments, and the other reviewer seems to have dug deeper into the proofs. The paper closes a gap in the literature of Bayesian regret by removing an assumption and proving a nice elliptical potential result.

**Key Questions For Authors:**

# Generalized elliptic potential lemma

It seems that in the end, you use the elliptic potential lemma for p = 1/2, like the other proofs. Could it be that the proof for any P case introduces some step that may not be tight? Could you provide some insight on why the result for any $p$ is interesting?


# Naive question on alternative proofs

Could you not have made a peeling on the norm of $\| \theta_\star \|$ and reused the previous proof on Thompson sampling that assumes bounded norm? If you call $\text{Reg}^{TS, B} (T)$ the regret of Thompson sampling when the norm is bounded by $B$ and horizon $T$ You can write something like :

$$ \text{Reg}^{TS} (T) \leq \text{Reg}^{TS, c(d , \Sigma_{0}) \log(T)} (T) +  \sum_{i=1}^{\infty} \mathbb{P}\Bigg( c(d , \Sigma_0) \log(T^{i+1}) \geq \| \theta_{\star} \| \geq    c(d , \Sigma_0) \log(T^{i}) \Bigg) c(d , \Sigma_0) \log(T^{i+1})  T  $$

And you could use Russo & Van Roy (2013) for $\text{Reg}^{TS, B} (T)$ and the fact that  $\mathbb{P}\left(  c(d , \Sigma_0) \log(T^{i+1}) \geq \| \theta_{\star} \| \geq   c(d , \Sigma_0) \log(T^{i}) \right) \leq \frac{1}{T^i}$.

Could that provide the same result as the first term is dealt with by previous work, and the sum in the second term converges and is a $\tilde{O}(c(d, \Sigma_0))$?

**Limitations:**

yes

**Strengths And Weaknesses:**

## Summary
I find the paper pretty clear on what it is trying to achieve and show. It provides numerous comparisons with other work, which I find very nice. (Even though I did not check all the subtleties of the other works.) I find the result interesting and useful, and it contributes to global understanding of algorithms by removing certain assumptions made by past authors and by accounting for other non-asymptotic terms (on the time horizon) that previous authors hid in the $O$ notation.


## Small suggestions (mainly some small polishing)

Line 79, second column, Line 95, second column, the cut in some math notation formulas (mainly due to the double column format).

Line 100, second column, Line 347. Maybe a smaller font size to make the constant on one line?

Some notation for "constant" should maybe include its dependence on the time horizon and dimension, like $\beta \rightarrow \beta(T,d)$ (line 114 column 2). Even though this is part of the standard proof scheme, I think it improves readability, especially for new bandit readers. And this is actually done for the final regret bound.

Line 230, I am not very fond of the use of  "—", and sometimes this creates paragraphs that are not very pleasing to the eye, line 373, second column.

---

> ### Author Rebuttal · Authors · 2026-03-25
>
> We thank the reviewer for the constructive review and clear feedback. We have fixed the wording and style issues accordingly.
>
> # Question on general elliptical potential lemma
>
> While we only use the lemma with $p=1/2$, the proof for the general case is just as clean as for the specific $p$. We included the proof for general $p$ because the technique is the same, and we did not want it to seem as though there was some idiosyncratic reason to use $p = 1/2$.
>
> With respect to the tightness, we showed on page 5 that the second term is tight up to $1/p$. It turns out that the first term is also tight up to universal constants. (See also the response to Reviewer bomQ)
>
> # Alternative proofs via Russo & Van Roy (2013)
>
> Thank you for pointing this out. We did not check all the details, but we think that such a proof would go through. Russo & Van Roy (2013) prove regret bounds like $\tilde{O}(\sigma d \sqrt{T} + dC)$, where rewards fall in $[0, C]$. To adapt their proof to our setting, we must choose $C$ to be on the order of $ r E |\theta_*| \succeq r \mathrm{Tr}(\Sigma_0)^{1/2}$. This yields a term on the order of $ d r \mathrm{Tr}(\Sigma_0)^{1/2}$. In contrast, our theorem 1 provides a term on the order of $ r \sqrt{d} \mathrm{Tr}(\Sigma_0^{1/2})$, which is always better by the Cauchy-Schwarz inequality.

---

> > ### Author Rebuttal · Reviewer_zKca · 2026-04-03
> >
> > I think the presentation is clear and the paper technically solid; I only had minor comments, and the other reviewer seems to have dug deeper into the proofs. The paper closes a gap in the literature of Bayesian regret by removing an assumption and proving a nice elliptical potential result.

---

### Decision · Program_Chairs · 2026-04-30

**Decision:**

Accept (regular)

**Comment:**

The paper studies Bayesian regret of Thompson Sampling in linear-Gaussian bandits and shows that the prior-dependent burn-in term can be decoupled additively from the leading regret term. The key contribution is a new generalized elliptical potential lemma, which enables this refinement and may be of independent interest.

Reviewers found the paper technically strong, clear, and well-positioned within the literature. The main strengths are the elegance and novelty of the analysis and the significance of closing an existing gap in regret bounds. Concerns were minor, mostly related to presentation details and limited discussion of broader implications.

The authors’ rebuttal addressed all questions satisfactorily, including strengthening the lower bound and clarifying technical points, and reviewers indicated their concerns were resolved.

Overall, given the strength of the contribution and positive consensus, I recommend acceptance, with minor revisions to improve clarity and presentation.